# Movie reconstruction from mouse visual cortex activity

**Joel Bauer[1,2]\*, Troy W Margrie[1†], Claudia Clopath[1,2†]**

[1]Sainsbury Wellcome Centre, University College London, London, United Kingdom; [2]Bioengineering Dept., Imperial College, London, United Kingdom

## eLife Assessment

This **valuable** study uses state-of-the-art neural encoding and video reconstruction methods to achieve a substantial improvement in video reconstruction quality from mouse neural data. It provides a **convincing** demonstration of how reconstruction performance can be improved by combining these methods. The goal of the study was improving reconstruction performance rather than advancing theoretical understanding of neural processing, so the results will be of practical interest to the brain decoding community.

**\*For correspondence:**
joel.bauer@ucl.ac.uk

[†]These authors contributed equally to this work

**Competing interest:** The authors declare that no competing interests exist.

**Abstract** The ability to reconstruct images represented by the brain has the potential to give us an intuitive understanding of what the brain sees. Reconstruction of visual input from human fMRI data has garnered significant attention in recent years. Comparatively less focus has been directed towards vision reconstruction from single-cell recordings, despite its potential to provide a more direct measure of the information represented by the brain. Here, we achieve high-quality reconstructions of natural movies presented to mice, from the activity of neurons in their visual cortex for the first time. Using our method of video optimization via backpropagation through a state-of-the-art dynamic neural encoding model, we reliably reconstruct 10 s movies at 30 Hz from two-photon calcium imaging data. We achieve a pixel-level correlation of 0.57 between ground-truth movies and single-trial reconstructions. Previous reconstructions based on awake mouse V1 neuronal responses to static images achieved a pixel-level correlation of 0.24 over a similar retinotopic area. We find that critical for high-quality reconstructions are the number of neurons in the dataset and the use of model ensembling. This paves the way for movie reconstruction to be used as a tool to investigate a variety of visual processing phenomena.

## Introduction

One fundamental aim of neuroscience is to eventually gain insight into the ongoing perceptual experience of humans and animals. Reconstruction of visual perception directly from brain activity has the potential to give us a deeper understanding of how the brain represents visual information. Over the past decade, there have been considerable advances in reconstructing images and videos from human brain activity (*Nishimoto et al., 2011*; *Shen et al., 2019a*; *Shen et al., 2019b*; *Rakhimberdina et al., 2021*; *Ren et al., 2021*; *Takagi and Nishimoto, 2023*; *Ozcelik and VanRullen, 2023*; *Ho et al., 2023*; *Scotti et al., 2023*; *Chen et al., 2023*; *Benchetrit et al., 2023*; *Kupershmidt et al., 2022*). These advances have largely leveraged deep learning techniques to interpret fMRI or MEG recordings, taking advantage of the fact that spatially separated clusters of neurons have distinct visual and semantic response properties (*Rakhimberdina et al., 2021*). Due to the low resolution of fMRI and MEG, relative to single neurons, the most successful models heavily rely on extracting semantic content and use diffusion models to generate semantically similar images and videos. Some

approaches combine low-level perceptual (retinotopic) and semantic information in separate modules to achieve even better image similarity (*Ren et al., 2021*; *Ozcelik and VanRullen, 2023*; *Scotti et al., 2023*). However, the pixel-level similarities are still relatively low. These methods are highly useful in humans, but their focus on semantic content may make them less useful when applied to non-human subjects or when using the reconstructed images to investigate visual processing.

Less attention has been given to image reconstruction from non-human brains. This is surprising given the advantages of large-scale single-cell-resolution recording techniques available in animal models, particularly mice. In the past, reconstructions using linear summation of receptive fields or Gabor filters have shown some success using responses from retinal ganglion cells (*Brackbill et al., 2020*), thalamo-cortical neurons in lateral geniculate nucleus (*Stanley et al., 1999*), and primary visual cortex (*Garasto et al., 2019*; *Yoshida and Ohki, 2020*). Recently, deep nonlinear neural networks have been used with promising results to reconstruct static images from mouse retina (*Zhang et al., 2020*; *Li et al., 2023*) and visual cortex (*Cobos et al., 2022*), and in particular from monkey V4 extracellular recordings (*Li et al., 2023*; *Pierzchlewicz et al., 2023*).

Here, we present a method for the reconstruction of 10 s movie clips using two-photon calcium imaging data recorded in mouse V1 (*Turishcheva et al., 2023*). Our method takes advantage of a state-of-the-art (SOTA) dynamic neural encoding model (DNEM) (*Baikulov, 2023a*) which predicts neuronal activity based on video input, as well as behavior. Our method allows us to successfully reconstruct videos despite the fact that V1 neuronal activity in awake mice is heavily modulated by behavioral factors, such as running speed (*Niell and Stryker, 2010*) and pupil diameter (correlated with arousal; *Reimer et al., 2014*). We then quantify the spatio-temporal limits of this reconstruction approach and identify key aspects of our method necessary for optimal performance.

## Results

### Video reconstruction using state-of-the-art dynamic neural encoding models

We used publicly available data provided by the Sensorium 2023 competition (*Turishcheva et al., 2023*; *Turishcheva et al., 2024*). The data included movies that were presented to mice and the evoked activity of V1 neurons along with pupil position, pupil diameter, and running speed. The neuronal activity was measured using two-photon imaging of GCaMP6s (*Chen et al., 2013*) fluorescence from 10 mice, with ≈8000 neurons from each mouse. In total, we reconstructed ten 10 s natural movies from 5 mice.

We used the winning model of the Sensorium 2023 competition which achieved a score of 0.301 (*Baikulov, 2023a*; *Turishcheva et al., 2024*) single-trial correlation between predicted and ground truth neuronal activity; (*Figure 1A*, *Figure 1—figure supplement 1B−C*). This state-of-the-art (SOTA) dynamic neural encoding model (DNEM), called DwiseNeuro was composed of three parts: core, cortex and readout. The model takes the video as input with the behavioral data (pupil position, pupil diameter, and running speed) broadcast to four additional channels of the video. This model achieved an average single-trial correlation between predicted and ground truth neural activity of 0.291 during the competition; this was later improved to 0.301. For context, the competition benchmark models achieved 0.106, 0.164, and 0.197 single-trial correlation, while the second and third place models achieved 0.265 and 0.243. Across the models entered in the competition, a variety of architectural components were used, including 2D and 3D convolutional layers, recurrent layers, and transformers, to name just a few. The original model weights of the winning model were not used to avoid reconstructing movies the model was trained on. Instead, we retrained 7 instances of the model using the same training data, which did not include the movies reserved for reconstruction. Beyond this point, the weights of the model were frozen, i.e., not influenced by future movie presentations.

To reconstruct the videos presented to mice, we iteratively optimized an initially blank input video to the SOTA DNEM until the predicted activity in response to this input matched the ground truth recorded neuronal activity. In effect, we optimized the input video to be perceptually similar with respect to the recorded neurons. To achieve this, we used an input optimization through gradient descent approach inspired by the optimization of maximally exciting images (*Walker et al., 2019*) and the reconstruction of static images (*Cobos et al., 2022*; *Pierzchlewicz et al., 2023*). The input videos were initialized as uniform gray values and the behavioral parameters (*Figure 1—figure supplement*

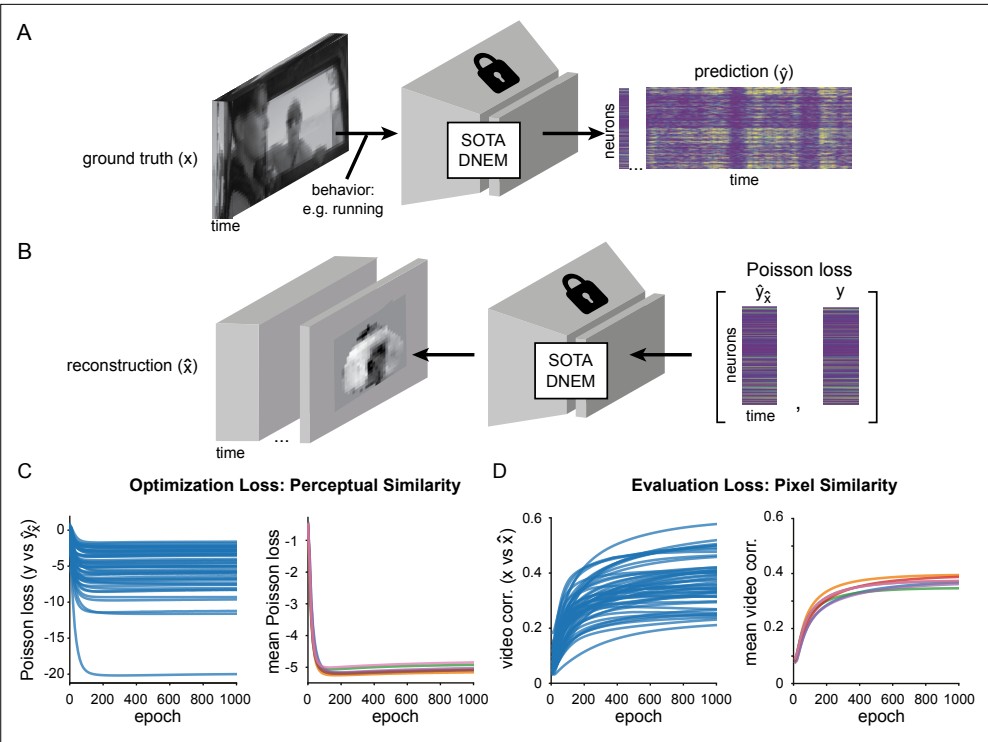

**Figure 1.** Video reconstruction from neuronal activity in mouse V1 (data provided by the Sensorium 2023 competition; *Turishcheva et al., 2023*; *Turishcheva et al., 2024*) using a state-of-the-art (SOTA) dynamic neural encoding model (DNEM; *Baikulov, 2023a*). (**A**) Dynamic neural encoding models (DNEMs) predict neuronal activity from mouse primary visual cortex, given a video and behavioral input. (**B**) We use a SOTA DNEM to reconstruct part of the input video given neuronal population activity, using gradient descent to optimize the input. (**C**) Poisson negative log likelihood loss across training steps between ground truth neuronal activity and predicted neuronal activity in response to reconstructed videos. Left: all 50 videos from 5 mice for one model. Right: average loss across all videos for seven model instances. (**D**) Spatio-temporal (pixel-by-pixel) correlation between reconstructed video and ground truth video.

The online version of this article includes the following source data and figure supplement(s) for figure 1:

**Source data 1.** Source data to *Figure 1*.

**Figure supplement 1.** Summary ethogram of state-of-the-art (SOTA) dynamic neural encoding model (DNEM) inputs, output predictions, and video reconstruction over time for three videos from three mice (same as *Figure 2A*).

**Figure supplement 1—source data 1.** Source data to *Figure 1—figure supplement 1*.

**Figure supplement 2.** Variations on the reconstruction method.

**Figure supplement 2—source data 1.** Source data to *Figure 1—figure supplement 2*.

**Figure supplement 3.** Receptive fields and transparency masks.

**Figure supplement 3—source data 1.** Source data to *Figure 1—figure supplement 3*.

---

*1A*) were added as additional channels, i.e., these were not reconstructed but given. The neuronal activity in response to the input video was predicted using the SOTA DNEM for a sliding window of 32 frames (1.067 s) with a stride of eight frames. We saw slightly better results with a stride of two frames, but in our case, this did not warrant the increase in training time. For each window, the difference between the predicted and ground truth responses was calculated, and this loss was backpropagated to the pixels of the input video to get the gradient of the loss with respect to each pixel. In effect, the input pixels were thus treated as if they were model weights. The gradients for each pixel were then averaged across all windows and the pixels of the input video updated accordingly (See Supplementary Algorithm 1).

The data from the Sensorium competition provided the activity of neurons within a 630 by 630 µm field of view for each mouse, i.e., covering roughly one-fifth of mouse V1. Due to the retinotopic

organization of V1 we, therefore, did not expect to get good reconstructions of the entire video frame. However, gradients still propagated to the full video frame and produced nonsensical results along the periphery of the video frames (*Figure 1—figure supplement 2B*). Inspired by previous work (*Mordvintsev et al., 2018*; *Willeke et al., 2026*) we, therefore, decided to apply a mask during training and evaluation. To generate these masks, we optimized a transparency layer placed at the input to the SOTA DNEM. High values are given to pixels that contribute to the accurate prediction of neuronal activity and represent the collective receptive field of the neural population. None of the reconstructed movies were used in the optimization of this transparency mask. The transparency masks are aligned with but not identical to the On-Off receptive field distribution maps using sparse noise (*Figure 1—figure supplement 3*). This mask was applied during the optimization of the reconstructed movies (training mask: binarized with threshold $\alpha = 0.5$) and applied again to the final reconstruction (evaluation mask: binarized with threshold $\alpha = 1$) (See Supplementary Algorithm 2). Applying the mask in two stages first boosts the performance of reconstruction itself and separately allows evaluation of the reconstruction in a region of high confidence, given the neural population available (*Figure 1—figure supplement 2*).

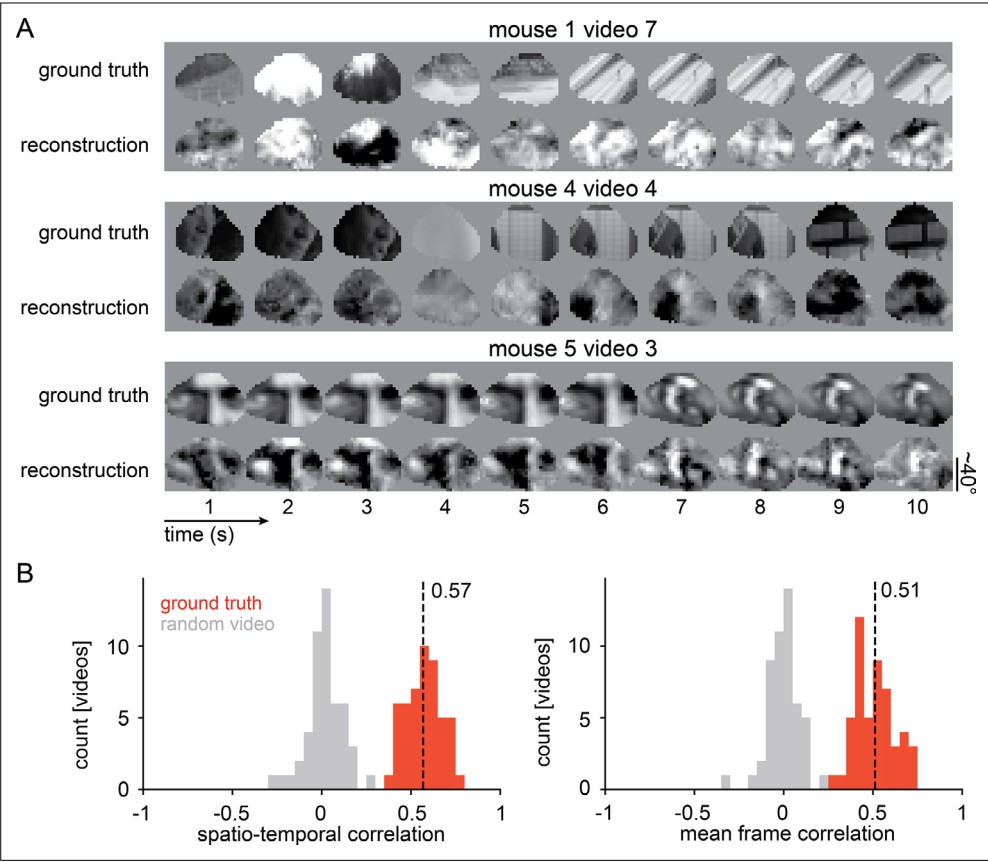

**Figure 2.** Reconstruction performance. (**A**) Three reconstructions of 10 s videos from different mice (see *Video 1* for the full set). Reconstructions have been luminance (mean pixel value across video) and contrast (standard deviation of pixel values across video) matched to ground truth. (**B**) The reconstructed videos have high correlation to ground truth in both spatio-temporal correlation (mean Pearson's correlation $r=0.569$ with 95% CIs 0.542–0.596, t-test between ground truth and random video $p=6.69 \times 10^{-49}$, n=50 videos from 5 mice) and mean frame correlation (mean Pearson's correlation $r=0.512$ with 95% CIs 0.481–0.543, t-test between ground truth and random video $p=4.29 \times 10^{-45}$, n=50 videos from 5 mice).

The online version of this article includes the following source data and figure supplement(s) for figure 2:

**Source data 1.** Source data to *Figure 2*.

**Figure supplement 1.** Reconstruction performance correlates with frame contrast but not with behavioral parameters.

**Figure supplement 1—source data 1.** Source data to *Figure 2—figure supplement 1*.

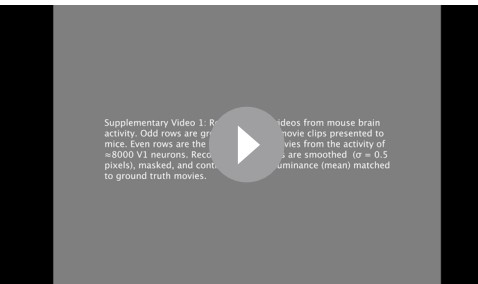

**Video 1.** Reconstructed natural videos from mouse brain activity. Odd rows are ground truth (GT) movie clips presented to mice. Even rows are the reconstructed movies from the activity of ≈8000 V1 neurons. Reconstructed movies are smoothed (σ=0.5 pixels), masked, and contrast (std) and luminance (mean) matched to ground truth movies.

https://elifesciences.org/articles/105081/figures#video1

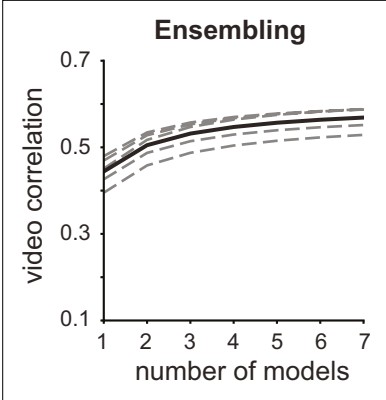

**Figure 3.** Model ensembling. Mean video correlation is improved when predictions from multiple models are averaged. Dashed lines are individual animals, and the solid line is the mean. One-way repeated measures ANOVA p=1.11 x 10$^{-16}$. Bonferroni-corrected paired t-test outcomes between consecutive ensemble sizes are all *p*<0.001, n=5 mice.

The online version of this article includes the following source data for figure 3:

**Source data 1.** Source data to *Figure 3*.

As the loss between predicted (*Figure 1—figure supplement 1D*) and ground truth responses (*Figure 1—figure supplement 1B*) decreased, the similarity between the reconstructed and ground truth input video increased (*Figure 1C–D*). We generated seven separate reconstructions from seven neural encoding models (trained on the same data) and averaged them. Finally, we applied a 3D Gaussian filter with sigma 0.5 pixels to reduce the remaining static noise (*Figure 1—figure supplement 2*) and applied the evaluation mask. When presenting videos in this paper, we normalize the mean and standard deviation of the reconstructions to the average and standard deviation of the corresponding ground truth movie before applying the evaluation masks, but this is not done for quantification except in *Figure 1—figure supplement 2D*. The Gaussian filter was not applied when evaluating spatial or temporal resolution (Figure 4, *Figure 4—figure supplement 1*, *Figure 4—figure supplement 2*).

## High-quality video reconstruction

As can be seen in *Figure 2* and *Video 1*, the reconstructed videos capture much of the spatial and temporal dynamics of the original input video. Because our optimization of the movies was based on a perceptual loss function, we were interested in how closely these movies matched the originals on the pixel level. To evaluate the performance of the video reconstructions we, therefore, correlated either all pixels from all time points between ground truth and reconstructed videos (Pearson's correlation *r*=0.569; to quantify temporal and spatial similarity), or the average correlation between all sets of frames (Pearson's correlation *r*=0.512; to quantify just spatial similarity) (*Figure 2B*). This represents a ≈2x higher pixel-level correlation over previous single-trial static image reconstructions from V1 in awake mice (image correlation 0.238+/-0.054 s.e.m. for awake mice) (*Yoshida and Ohki, 2020*) over a similar retinotopic area (≈43°×43°) while also capturing temporal dynamics. However, we would like to stress that directly comparing static image reconstruction methods with movie reconstruction approaches is fundamentally problematic, as they rely on different data types both during training and evaluation (temporally averaged vs continuous neural activity, images flashed at fixed intervals vs continuous movies).

Reconstruction quality, however, was not consistent across movies (*Figure 2B*) or constant throughout the 10 s videos (*Figure 1—figure supplement 1E*). We, therefore, investigated what factors may cause these fluctuations by correlating video motion energy, contrast, and luminance, as well as running speed, pupil diameter and eye movement with frame correlation. We found that contrast correlated with frame correlation, but only to a moderate degree. Video motion energy shows a trend but was not significant (*Figure 2—figure supplement 1A*). We also found that the ability of the SOTA DNEM to predict neural activity correlated with reconstruction performance. This

could be because some frames are harder to reconstruct due to their content (high temporal and spatial frequencies) or because neural activity in these moments was influenced by factors that the model cannot take into account.

## Ensembling

We found that the seven instances of the SOTA DNEMs by themselves performed similarly in terms of reconstructed video correlation (*Figure 1D*), but that this correlation was significantly increased by taking the average across reconstructions from different models (*Figure 3*) – A technique known as bagging, and more generally ensembling (*Breiman, 1996*). We averaged over seven model instances, which gave a performance increase of 28.0%, but the largest gain in performance, 13.7%, came from averaging across just 2 models (*Figure 3*). Doubling the number of models to four increased the performance by another 8.32%. Individual models produced reconstructions with high-frequency noise in the temporal and spatial domains. We, therefore, think the increase in performance from ensembling is mostly an effect of averaging out this high-frequency noise. On the other hand, it is possible that averaging over separately optimized reconstructions degrades high-frequency information. We, therefore, tested whether averaging pixel gradients from all models at each iteration rather than averaging the final movies yields higher performance, but we observed no improvement (*Figure 1—figure supplement 2C*). Overall, although ensembling over models trained on separate data splits is a computationally expensive method, it substantially improved reconstruction quality.

## Not all spatial and temporal frequencies are reconstructed equally

While the reconstructed videos achieve high correlation to ground truth, it is not entirely clear if the remaining deviations are due to the limitations of the model or arise from the recorded neurons themselves. To assess the resolution limits of our reconstruction process, we assessed the model's ability to reconstruct synthetic stimuli at varying spatial and temporal resolutions in a noise-free scenario.

To quantify which spatial and temporal frequencies our reconstruction approach is able to capture, we used a Gaussian noise stimulus set generated using a Gaussian process (https://github.com/TomGeorge1234/gp_video; *George, 2024*; *Figure 4*). The dataset consisted of 49, 2 s, 36 by 36 pixel videos at 30 Hz, which varied in the spatial and temporal length constants. As we did not have ground truth neuronal activity in response to this stimulus set, we first predicted the neuronal responses given these videos using the ensembled SOTA DNEMs. We then used gradient descent to reconstruct the original input using these predicted neuronal responses as the target. In this way, we generated reconstructions in an ideal case with no biological noise and assuming the SOTA DNEM perfectly predicts neuronal activity (*Figure 4B* and *Video 2*). This means the video reconstruction quality loss reflects the inefficiency of the reconstruction process itself without the additional loss or transformation of information by processes, such as top-down modulation, e.g., predictive coding or selective feature attention (see Discussion). We found that the reconstruction process failed at high spatial frequencies (<1 pixel, or <3.4° retinotopy) and performed worse at high temporal frequencies (<1 frame, or >30 Hz) (*Figure 4C* and *Video 2*). We repeated this analysis using full-field high-contrast square gratings drifting in the four cardinal directions and similarly found that high spatial and temporal frequencies were not reconstructed as well as low-spatial and temporal frequency gratings (*Figure 4—figure supplement 1* and *Video 3*). We also found that beyond the spatial reconstruction limit, the reconstructions from phase-inverted Gaussian noise stimuli had higher correlation with each other than with their ground truth stimuli (*Figure 4D*). Nevertheless, even when the reconstructions were not captured on the pixel level, they did capture some of the spatial entropy and motion energy of the ground truth stimuli (*Figure 4—figure supplement 2A–B*).

To test if model ensembling improves Gaussian noise reconstruction quality across all spatial and temporal length constants uniformly, we subtracted the average video correlation across the seven model instances from the video correlation of the average video (i.e. ensembled video reconstruction quality minus unensembled video reconstruction quality; *Figure 4—figure supplement 2C*). We found that, in particular, short temporal and spatial length constant stimuli improved in correlation, supporting our hypothesis that ensembling mitigates the high-frequency noise we observed in the reconstruction from individual models.

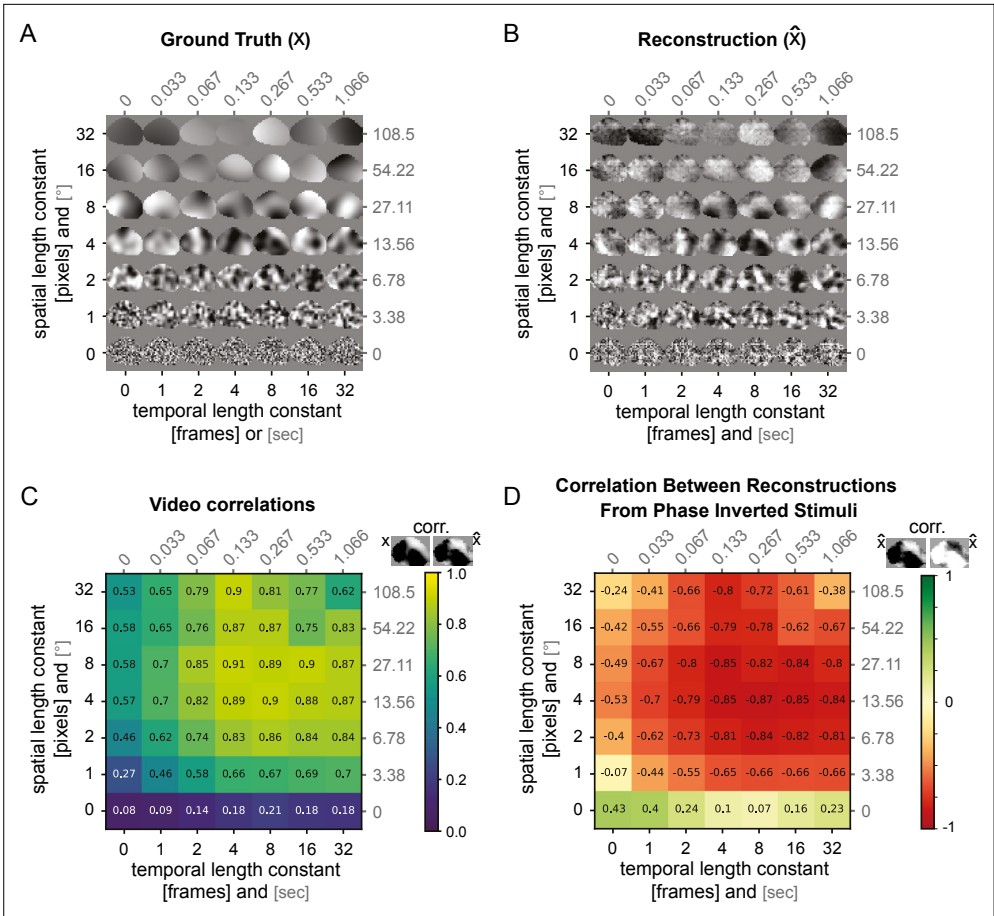

**Figure 4.** Reconstruction of Gaussian noise across the spatial and temporal spectrum using predicted activity. (**A**) Example Gaussian noise stimulus set with evaluation mask for one mouse. Shown is the last frame of a 2 s video. (**B**) Reconstructed Gaussian stimuli with state-of-the-art (SOTA) dynamic neural encoding model (DNEM) predicted neuronal activity as the target (see also *Video 2*). (**C**) Video correlation between ground truth (**A**) and reconstructed videos (**B**) across the range of spatial and temporal length constants. For each stimulus type, the average correlation across five movies reconstructed from the SOTA DNEM of 3 mice is given. (**D**) Video correlation between reconstructions from phase-inverted Gaussian noise stimuli.

The online version of this article includes the following source data and figure supplement(s) for figure 4:

**Figure supplement 1.** Reconstruction of drifting grating stimuli with different spatial and temporal frequencies using predicted activity.

**Figure supplement 1—source data 1.** Source data to *Figure 4—figure supplement 1*.

**Figure supplement 2.** Gaussian Noise reconstruction: shannon entropy, motion energy .

**Figure supplement 2—source data 1.** Source data to *Figure 1—figure supplement 2*.

## Neuronal population size

In order to design future *in vivo* experiments to investigate visual processing using our video reconstruction approach, it would be useful to know how reconstruction performance scales with the number of recorded neurons. This is vital for prioritizing experimental parameters, such as weighing between sampling density within a similar retinotopic area and retinotopic coverage to maximize both video reconstruction quality and visual coverage. We, therefore, performed an *in silico* ablation experiment, dropping either 50%, 75%, or 87.5% of the total recorded population of ≈8000 neurons per mouse by setting their activity to 0 (*Figure 5*). We found that dropping 50% of the neurons reduced the video correlation by only 9.96% while dropping 75% reduced the performance by 24.9%. We would, therefore, argue that ≈4000–8000 neurons within a 630 by 630 μm area (≈10000–20000 neurons/mm²) of mouse V1 would provide a balance when compromising between density and 2D coverage.

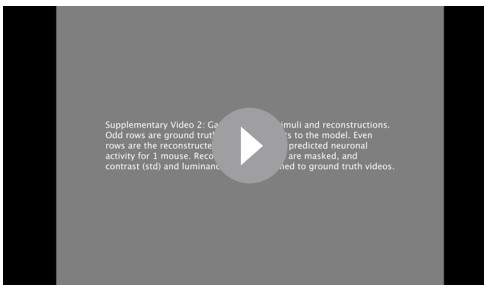

**Video 2.** Gaussian noise stimuli and reconstructions. Odd rows are ground truth (GT) video inputs to the model. Even rows are the reconstructed videos from the predicted neuronal activity for 1one mouse. Reconstructed movies are masked, and contrast (std) and luminance (mean) matched to ground truth videos. https://elifesciences.org/articles/105081/figures#video2

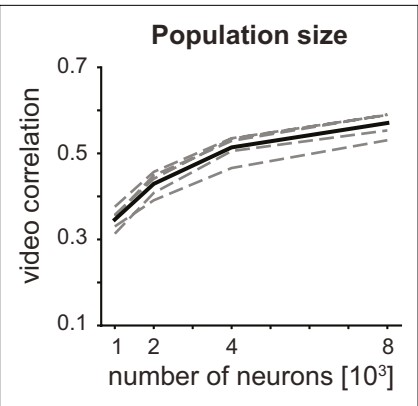

**Figure 5.** Video reconstruction using fewer neurons (i.e. population ablation) leads to lower reconstruction quality. Dashed lines are individual animals, and the solid line is the mean. One-way repeated measures ANOVA p=5.70 x 10⁻¹³. Bonferroni corrected paired t-test outcomes between consecutive drops in population size are all *p*<0.001, n=5 mice.

The online version of this article includes the following source data for figure 5:

**Source data 1.** Source data to *Figure 5*.

## Visualization of reconstruction error

One advantage of stimulus reconstruction compared to stimulus identity decoding (i.e. classification) is that it is possible to visualize the deviation of the reconstructed stimuli from what is expected. This is interesting because reconstruction performance is not stable over time but fluctuates (*Figure 6A*), likely due to the fact that the DNEM does not have access to all possible factors that influence neural activity. When using our reconstruction method, it is not the input stimulus similarity that is optimized, but the evoked activity of the stimulus. As a consequence, the predicted neural response from the reconstructed movie is more similar to the experimental neural response compared to the predicted neural response evoked by the original ground truth movie (*Figure 6B*). It is possible to visualize this deviation on a pixel level by subtracting the experimentally derived movie reconstruction (i.e. based on measured neural responses) from the *in silico* simulation derived movie reconstruction (i.e. first predict activity based on the ground truth video and then reconstruct the movie based on the resulting simulated neural activity) (*Figure 6* and *Video 4*). With the current dataset, it is not possible to test if these deviations reflect failures of the encoding model to predict neural activity given the sensory stimulus or true deviations of the images represented by the neural population from the sensory stimulus, but this approach may be an interesting method for investigating when and why model predictions of neural activity deviate from the experimentally measured activity.

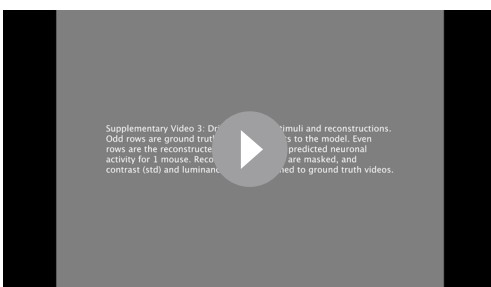

**Video 3.** Drifting grating stimuli and reconstructions. Odd rows are ground truth (GT) video inputs to the model. Even rows are the reconstructed videos from the predicted neuronal activity for 1one mouse. Reconstructed movies are masked, and contrast (std) and luminance (mean) matched to ground truth videos. https://elifesciences.org/articles/105081/figures#video3

## Discussion
### Stimulus identification vs reconstruction

Stimulus identification, i.e., identifying the most likely stimulus from a constrained set, has been a popular approach for quantifying whether a population of neurons encodes the identity of a particular stimulus (*Földiák, 1993*, *Kay et al., 2008*). This approach has, for instance, been used to decode frame identity within a movie (*Deitch et al., 2021*; *Xia et al., 2021*; *Schneider et al., 2023*; *Chen et al., 2024*). Some of these approaches have also been used to reorder the frames of the ground truth movie (*Schneider*

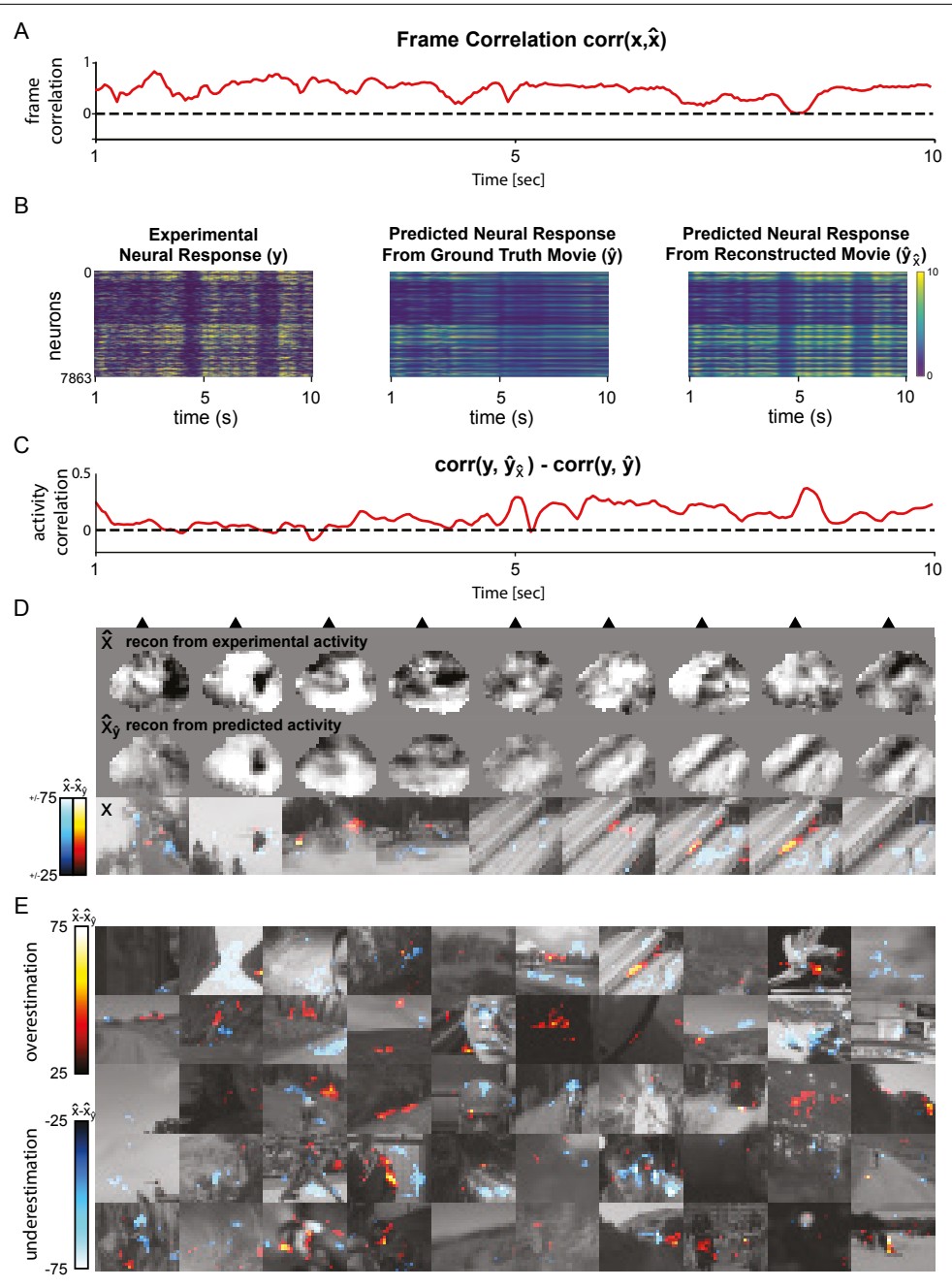

**Figure 6.** Comparison of reconstructions from experimental responses vs expected responses and their visualization as error maps. (**A**) Frame-by-frame correlation between reconstructed and ground truth video for mouse 1 trial 7 (same as *Figure 1—figure supplement 1E*). (**B**) From left to right: experimental (ground truth) neural activity $y$, neural activity predicted by dynamic neural encoding model (DNEM) from ground truth video $\hat{y}$, neural activity predicted by DNEM based on reconstructed movie $\hat{y}_{\hat{x}}$. (**C**) Difference between the correlation of true neural response $y$ with predicted neural response from the ground truth movie $\hat{y}$, and the correlation of true neural response $y$ with predicted neural response from the reconstructed movie $\hat{y}_{\hat{x}}$. (**D**) 9 frames from mouse 1 trial 7. From top to bottom: reconstructed movie $\hat{x}$, reconstructed movie from predicted neural response to ground truth movie $\hat{x}_{\hat{y}}$, ground truth movie $x$ with overlayed heatmap of the difference between $\hat{x}$ and $\hat{x}_{\hat{y}}$ (error map). (**E**) Error map of one frame from all 50 movie clips. Each row is 10 trials from one mouse. See also *Video 4* .

The online version of this article includes the following source data for figure 6:

**Source data 1.** Source data to *Figure 6*.

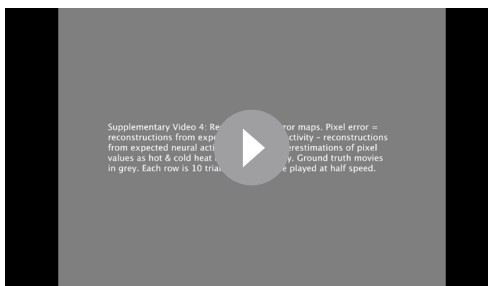

**Video 4.** Reconstruction error maps. Pixel error = reconstructions from experimental neural activity – reconstructions from expected neural activity. Over-&and underestimations of pixel values as hot &and cold heat maps, respectively. Ground truth movies in gray. Each row is 10 trials from 1one mouse played at half speed.

https://elifesciences.org/articles/105081/figures#video4

et al., 2023) based on the decoded frame identity. Importantly, stimulus identification methods are distinct from stimulus reconstruction, where the aim is to recreate what the sensory content of a neuronal code is in a way that generalizes to new sensory stimuli (*Rakhimberdina et al., 2021*). This is inherently a more demanding task because the range of possible solutions is much larger. Although stimulus identification is a valuable tool for understanding the information content of a population code, stimulus reconstruction could provide a more generalizable approach, because it can be applied to novel stimuli.

## Comparison to other reconstruction methods

There has recently been a growing number of publications in the field of image reconstruction, primarily from fMRI data, and a comprehensive review of all the approaches is outside the scope of this paper. However, we will briefly summarize the most common approaches and how they relate to our own method. In general, image reconstruction methods can be categorized into one of four groups: direct decoding models, encoder-decoder models, invertible encoding models, and encoder model input optimization.

Direct decoders directly decode the input image/videos from neuronal activity with deep neuronal networks (*Shen et al., 2019a*; *Zhang et al., 2020*; *Li et al., 2023*). When training direct decoders, the decoders can be pretrained (*Ren et al., 2021*) or additional constraints can be added to the loss function to encourage the decoder to produce images that adhere to learned image statistics (*Shen et al., 2019a*; *Kupershmidt et al., 2022*). A direct decoder approach has been used for video reconstruction in mice (*Chen et al., 2024*), but in that case, the training and test movies were the same, meaning it is unclear if out-of-training set generalization was achieved (a key distinction between sensory reconstruction and stimulus identification, see previous section).

In encoder-decoder models, the aim is to combine separately trained brain encoders (brain activity to latent space) and decoders (latent space to image/video). Recently, this approach has become particularly popular because it allows the use of SOTA generative image models, such as stable diffusion (*Rombach et al., 2021*; *Takagi and Nishimoto, 2023*; *Scotti et al., 2023*; *Chen et al., 2023*; *Benchetrit et al., 2023*). The encoder part of the models are first trained to translate brain activity into a latent space that the pretrained generative networks can interpret. Because these latent spaces are often conditioned on semantic information, this lends itself to separate processing of low-level visual and high-level semantic information from brain activity (*Scotti et al., 2023*).

Invertible encoding models are encoding models which, once trained to predict neuronal activity, can implicitly be inverted to predict sensory input given brain activity. We would also include those models in this class which first compute the receptive field or preferred stimulus of neurons (or voxels) and reconstruct the input as the weighted sum of the receptive fields by their activity (*Stanley et al., 1999*; *Thirion et al., 2006*; *Garasto et al., 2019*; *Brackbill et al., 2020*; *Yoshida and Ohki, 2020*; *Nishimoto et al., 2011*). The downside of this approach is that invertible linear models generally underperform in terms of capturing the coding properties of neurons compared to more complex deep neural networks (*Willeke et al., 2023*).

Encoder input optimization also involves first training an encoder which predicts the activity of neurons or voxels given sensory input. Once trained, the encoder is fixed, and the input to the network is optimized using backpropagation until the predicted activity matches the observed activity (*Pierzchlewicz et al., 2023*). Unlike with invertible encoding models, any SOTA neuronal encoding model can be used. But like invertible models, the networks are not specifically trained to reconstruct images, so they may be less likely to extrapolate information encoded by the brain by learning general image statistics. There is some evidence to support this, static image reconstructions which were

optimized to evoke similar *in silico* predicted neural activity also evoke more similar neural responses *in vivo* when compared to other methods that optimized image similarity directly (*Cobos et al., 2022*).

Although outlined here as four distinct classes, these approaches can be combined. For instance, encoder input optimization can be combined with image diffusion (*Pierzchlewicz et al., 2023*) and in principle, invertible models could also be combined in such a way.

We chose to pursue a pure encoder input optimization approach for single-cell mouse visual cortex activity for two reasons. First, there have been considerable advances in the performance of neuronal encoding models for dynamic visual stimuli (*Sinz et al., 2018*; *Wang et al., 2025*; *Turishcheva et al., 2024*) and we aimed to take advantage of these developments. Second, the addition of a generative decoder trained to produce high-quality images brings with it the risk of extrapolating information based on general image statistics rather than interpreting what the brain is representing. In some cases, the brain may not be encoding coherent images, and in those cases, we would argue image reconstruction should fail, rather than producing an image when only the semantic information is present.

## Key contributions and limitations

We demonstrate high-quality video reconstruction from mouse V1 using SOTA DNEMs to iteratively optimize the input video to match the resulting predicted activity with the recorded neuronal activity. Key to achieving high-quality reconstructions is model ensembling and using a large enough number of recorded neurons over a given retinotopic area.

While we averaged the video reconstructions from several models, an alternative method would be to average the gradients calculated by multiple models at each epoch, as has been done for the generation of maximally exciting images in the past (*Walker et al., 2019*). When using video models, this can be an impractical solution due to the amount of GPU memory required, but in principle, there might be situations in which averaging gradients yields better reconstructions. For instance, there may be multiple solutions for the activation pattern of a neural population, e.g., if their responses are translation/phase invariant (*Ito et al., 1995*; *Tacchetti et al., 2018*). In such a case, averaging 'misaligned' reconstructions from multiple models might degrade overall quality. However, we observed no performance improvement when ensembling with gradients instead of ensembling with reconstructions.

The SOTA DNEM we used takes video data at an angular resolution of 3.4°/pixels at the center of the screen which is about 3x worse than the visual acuity of mice (≈0.5 cycles/° *Prusky and Douglas, 2004*). As our model can reconstruct Gaussian noise stimuli down to a spatial length constant of 1 pixel, and drifting gratings up to a spatial frequency of 0.071 cycles/°, there is still some potential for improving spatial resolution. To close this gap and achieve reconstructions equivalent to the limit of mouse visual acuity, a different dataset and model would likely need to be developed. However, the frame rate of the videos the SOTA DNEM takes as input (30 Hz) is faster than the flicker fusion frequency of mice (14 Hz; *Nomura et al., 2019*) and our tests with Gaussian noise and drifting grating stimuli show that the temporal resolution of reconstruction is close to this expected limit. Future efforts should, therefore, focus on the spatial resolution of video reconstruction rather than the temporal resolution.

It is, however, unclear how closely the representation of vision by the brain is expected to match the actual input. There are a number of visual processing phenomena that have previously been identified, which leads us to suspect that some deviations between video reconstructions and ground truth input are to be expected. One such phenomenon is predictive coding (*Rao and Ballard, 1999*; *Fiser et al., 2016*). It is possible that the unexpected parts of visual stimuli are sharper and have higher contrast compared to the expected parts when reconstructed from neuronal activity. Alternatively, perceptual learning is a phenomenon where visual stimulus detection or discriminability is enhanced through prolonged training (*Li, 2016*) and is associated with changes in the tuning distribution of neurons in the visual system (*Goltstein et al., 2013*; *Poort et al., 2015*; *Jurjut et al., 2017*; *Schumacher et al., 2022*). Similarly, selective feature attention can modulate the response amplitude of neurons that have a preference for the features that are currently being attended to *Kanamori and Mrsic-Flogel, 2022*. Visual task engagement and training could, therefore, alter the accuracy and biases of what features of a video can accurately be reconstructed from the neuronal activity. Visualizing differences between movie reconstructions from experimentally derived recordings to those from predicted activity, as we have done, may be an interesting approach.

Although fMRI-based reconstruction techniques are starting to be used to investigate visual phenomena in humans (such as illusions *Cheng et al., 2023* and mental imagery *Shen et al., 2019b*, *Koide-Majima et al., 2024*, *Kalantari et al., 2025*), visual processing phenomena are likely difficult to investigate using existing fMRI-based reconstruction approaches, due to the low spatial and temporal resolution of the data. Additionally, many of these fMRI-based reconstruction approaches rely on the use of pretrained generative diffusion models to achieve more naturalistic and semantically inter-pretable images (*Takagi and Nishimoto, 2023*; *Ozcelik and VanRullen, 2023*; *Scotti et al., 2023*; *Chen et al., 2023*), but very likely at the cost of introducing information that may not be present in the actual neuronal representation. In contrast, our video reconstruction approach using single-trial single-cell resolution recordings, without a pretrained generative model, provides a more accurate method to investigate visual processing phenomena, such as predictive coding, perceptual learning, and selective feature attention.

**In conclusion**, we reconstruct videos presented to mice based on single-trial activity of neurons in the mouse visual cortex. This paves the way to using movie reconstruction as a tool to investigate a variety of visual processing phenomena.

## Methods

### Source data

The data was provided by the Sensorium 2023 competition (*Turishcheva et al., 2023*; *Turishcheva et al., 2024*) and downloaded from https://gin.g-node.org/pollytur/Sensorium2023Data and https://gin.g-node.org/pollytur/sensorium_2023_dataset. The data included grayscale movies presented to the mice at 30 Hz on a 31.8 by 56.5 cm monitor 15 cm from and perpendicular to the left eye. The movies were provided as spatially downsampled versions of the original screen resolution to 36 by 64 pixels, corresponding to an angular resolution of 3.4°/pixel at the center of the screen. The pupil position and diameter were recorded at 20 Hz and the running at 100 Hz. The neuronal activity was measured using two-photon imaging (*Denk et al., 1990*) of GCaMP6s (*Chen et al., 2013*) fluores-cence at 8 Hz, extracted and deconvolved using the CAIMAN pipeline (*Giovannucci et al., 2019*). For each of the 10 mice, the activity of ≈8000 neurons was provided. The different data types were resampled to 30 Hz.

### State-of-the-art dynamic neural encoding model

We used the winning model of the Sensorium 2023 competition, DwiseNeuro (*Turishcheva et al., 2023*; *Turishcheva et al., 2024*). The code for the SOTA DNEM was downloaded from https://github.com/lRomul/sensorium (*Baikulov, 2023b*). The winning model consists of 3 main components: core, cortex, and readout. The core largely consisted of factorized 3D convolution blocks with residual connections, positional encoding (*Vaswani et al., 2017*), and SiLU activations (*Elfwing et al., 2018*) followed by spatial average pooling. The cortex consisted of three fully connected layers. The readout consisted of a 1D convolution for each mouse with a final Softplus nonlinearity that gives activity predictions for all neurons of each mouse. The kernel of the input layer had size 16 with a dilation of 2 in the time dimension, so spanned 32 video frames.

The original ensemble of models consisted of 7 model instances trained on a sevenfold cross-validation split of all available Sensorium 2023 competition data (≈1 hr of training data and ≈8 min of cross-validation data per fold from each mouse). Each model instance was trained on 6 of 7 data folds, with different validation data excluded from training for each model. To allow ensembled recon-structions of videos without test set contamination, we instead retrained the models with a shared validation fold, i.e., we retrained the models leaving out the same validation data for all seven model instances. The only other difference in the training procedure was that we retrained the models using a batch size of 24 instead of 32. This did not change the performance of neuronal response prediction on the withheld data folds (mean validation fold predicted vs ground truth response correlation for original weights: 0.293; and retrained weights: 0.291). We also did not use model distillation, while the original model did (see https://github.com/lRomul/sensorium; *Baikulov, 2023b*).

We chose the first 10 movies in data fold 0 (assigned as part of the DNEM code using a video hashing function) for reconstructions. We additionally excluded nine movies which were incorrectly assigned to fold 0 and replaced them with other movie clips from fold 0.

## Additional visual stimuli

The Gaussian noise stimuli were downloaded from https://github.com/TomGeorge1234/gp_video (*George, 2024*) and spanned a range of 0–32 pixels in spatial length constant and 0–32 frames in temporal length constant used in the Gaussian process. 5 separately generated movies of 2 s each were generated and combined with their phase-inverted versions to give a total of 10 trials.

The drifting grating stimuli were produced using PsychoPy (*Peirce et al., 2019*) and ranged from 0.5 to 0.062 cycles/degree and 0.5–0 cycles/s, with 2 s of movie for each cardinal direction. These ranges were chosen to avoid aliasing effects in the 36 by 64 pixel videos. The highest temporal frequency corresponds to a flicker stimulus.

The receptive field mapping stimulus, i.e., sparse noise stimulus, consisted of a pre-stimulus gray (gray value 127) screen period of 0.5 s, a 0.5 s stimulus period where one pixel was set to either 0 (Off) or 255 (On), and a 0.5 s post-stimulus gray screen period. The full stimulus set consisted of 4608 stimuli, one On and one Off stimulus for every pixel of the 36 by 64 movie.

## Mask training

To generate the transparency masks, we used an alpha blending approach inspired by *Mordvintsev et al., 2018* and *Willeke et al., 2026*. A transparency layer was placed at the input to the SOTA DNEM. This transparency layer was used to alpha blend the true video $V$ with another randomly selected background video $BG$ from the data:

$$V_{BG} = V * \alpha + BG * (1 - \alpha) \tag{1}$$

where $\alpha$ is the 2D transparency mask and $V_{BG}$ is the blended input video. This mask was optimized using stochastic gradient descent (for 1000 epochs with learning rate 10) with mean squared error ($MSE$) loss between the true responses $y$ and the predicted responses $\hat{y}$ scaled by the average weight of the transparency mask $\bar{\alpha}$:

$$MSE(y, \hat{y}) = \frac{1}{n} \sum_{i=1}^{n} (y - \hat{y})^2 \tag{2}$$

$$Loss = \mathrm{MSE}(y, \hat{y} * (1 - \bar{\alpha})) \tag{3}$$

where $n$ is the total number of neurons. The mask was initialized as uniform noise between 0 and 0.05. At each epoch, the neuronal activity in response to a randomly selected 32-frame video segment from the training set was predicted and the gradients of the loss (*Equation 3*) with respect to the pixels in the transparency mask $\alpha$ were calculated for each video frame. The gradients were normalized by their matrix norm, clipped to between –1 and 1, and averaged across frames. The gradients were smoothed with a 2D Gaussian kernel of σ = 5 and subtracted from the transparency mask. The transparency mask was only calculated using one SOTA DNEM instance using its validation fold. See Supplementary Algorithm 2.

The transparency mask was thresholded and binarized at 0.5 for the masked gradients $\nabla_{masked}$ or 1 for the masked videos for evaluation $V_{eval}$:

$$\nabla_{masked} = \nabla * (\alpha > 0.5) \tag{4}$$

$$V_{eval} = V * (\alpha \geq 1) \tag{5}$$

where $\nabla$ is the gradients of the loss with respect to each pixel in the video and $V$ is the reconstructed video before masking. These masks were trained independently for each mouse using one model instance with the original weights of the model https://github.com/lRomul/sensorium (*Baikulov, 2023b*), not the retrained models used in the rest of this paper to reconstruct the videos.

## Video reconstruction

To reconstruct the input video, we initialized the video as uniform gray values and concatenated the ground truth behavioral parameters. The SOTA DNEM took 32 frames at a time, and we shifted this window by eight frames until all frames of the whole 10 s video were covered. For each 32-frame window, the Poisson negative log-likelihood loss between the predicted and true neuronal responses was calculated:

$$Loss(y, \hat{y}) = \sum \hat{y}_{neuron} - y_{neuron} \log(\hat{y}_{neuron} + 10^{-8}) \tag{6}$$

where $\hat{y}$ are the predicted responses and $y$ are the ground truth responses. The gradients of the loss with respect to each pixel of the input video were calculated for each window of frames and averaged across all windows. The gradients for each pixel were normalized by the matrix norm across all gradients and clipped to between –1 and 1. The gradients were masked (*Equation 4*) and applied to the input video using Adam (ß1 = 0.9) without second-order momentum (*Kingma and Ba, 2014*) for 1000 epochs and a learning rate of 1000, with a learning rate warm-up for the first 10 epochs. After each epoch, the video was clipped to between 0 and 255. The optimization was run for 1000 epochs. Seven reconstructions from seven model instances were averaged, denoised with a 3D Gaussian filter σ = 0.5 (unless specified otherwise), and masked with the evaluation mask. See Supplementary Algorithm 1. Optimizing each 10 s video with one model instance for 1000 epochs took ≈60 min using a desktop with an RTX4070 GPU.

## Reconstruction quality assessment

To evaluate the similarity between reconstructed and ground truth videos, we used the mean Pearson's correlation between pixels of corresponding frames to evaluate spatial similarity:

$$\text{mean frame correlation} = \frac{1}{f} \sum_{i=1}^{f} \frac{cov(x_i, \hat{x}_i)}{\sigma_{x_i} \sigma_{\hat{x}_i}} \tag{7}$$

where $f$ is the number of frames, and $x_i$ and $\hat{x}_i$ are the ground truth and reconstructed frames. To evaluate temporal and spatial similarity between ground truth and reconstructed videos, we used the Pearson's correlation between all pixels of the whole movie:

$$\text{video correlation} = \frac{cov(x, \hat{x})}{\sigma_x \sigma_{\hat{x}}} \tag{8}$$

To calculate the Shannon entropy, we first computed the intensity histogram of the pixels inside the evaluation mask for every frame (25 bins between 0 and 255). Shannon entropy of one frame ($H_f$) was then calculated as:

$$H_f = - \sum_{k=1}^{n} p_k \log_2 p_k \tag{9}$$

where $p_k$ is the normalized histogram count of bin $k$ (only including non-zero bins). For each movie, the average Shannon entropy across frames is taken. $n$ is the total number of non-zero bins.

The motion energy of a frame ($E_f$) is calculated as:

$$E_f = \frac{1}{n} \sum_{i=1}^{n} \left| V_{f,i} - V_{f-1,i} \right| \tag{10}$$

where $V_{f,i}$ is the intensity value for one pixel $i$ inside the evaluation mask at frame $f$, $n$ is the total number of pixels inside the mask.

## Retinotopic mapping

To calculate the receptive fields of neurons *in silico*, we predicted each neuron's response to the full sparse noise stimulus set using the ensembles' prediction of seven SOTA DNEM instances. The response map across pixels for each neuron ($OnR_{h,w,n}$) was defined as:

$$OnR_{h,w,n} = \frac{(R_{stim} - R_{pre})}{R_{pre}} \tag{11}$$

where h and w denote the position of the pixel on the screen, and $n$ the neuron. $R_{stim}$ is the predicted response of the neuron during the stimulus period and $R_{pre}$ during the pre-stimulus period. $OnR_{h,w,n}$ was thresholded at 0.1. The same procedure was done to calculate $OffR_{h,w,n}$. The $OnR$ and $OffR$ maps were smoothed using a 2D Gaussian filter with σ = 2 and then normalized by the maximum value for

each neuron. The On and Off receptive field centers were defined as the pixel with the maximum value for each neuron. We calculate the On-Off receptive fields, for example, neurons as:

$$\text{On-Off receptive fields} = \frac{OnR_n - OffR_n}{Max(OnR_n - OffR_n)} \tag{12}$$

and calculate the population On or Off response as:

$$\text{population On or Off response} = \frac{OnR + OffR}{Max(|OnR + OffR|)} \tag{13}$$

## Reconstruction area calculation

To calculate the retinotopic diameter of a mask, we first computed the retinotopic area of each pixel of the movie based on the screen size (31.8 cm by 56.5 cm) and distance from the mouse eye (15 cm). Strictly speaking, this is the visuotopic area as it does not take eye position into account, but we refer to it as retinotopic for simplicity. We then take the sum of all pixel areas for an evaluation mask with a given $\alpha$ threshold. Then we define the retinotopic diameter of this area ($A$) as:

$$\text{retinotopic diameter} = 2\sqrt{\frac{A}{\pi}} \tag{14}$$

## Error map calculation

To calculate the error maps, we reconstruct movie clips either from the experimental neural responses or the predicted neural responses given the ground truth movie and took the difference:

$$\text{positive error} = \hat{x} - \hat{x}_{\hat{y}} > 0 \tag{15}$$

$$\text{negative error} = \hat{x} - \hat{x}_{\hat{y}} < 0 \tag{16}$$

where $x$ is the ground truth video, $y$ is the experimental neural activity, $\hat{x}$ is the reconstructed movie from $y$, $\hat{y}_{\hat{x}}$ is the predicted neural activity from $\hat{x}$, $\hat{y}$ is the predicted neural activity from $x$, and $\hat{x}_{\hat{y}}$ is the reconstructed movie from $\hat{y}$. Using Fiji (*Schindelin et al., 2012*), the positive error (LUT: red hot, range 25–75), negative error (LUT: cyan hot, range –25 to –75), and ground truth video (LUT: gray scale, range 0–255) were then combined into a composite image.

## Acknowledgements

We would like to thank Emmanuel Bauer, Sandra Reinert, and the anonymous reviewers for useful input and discussions, and Tom George for the Gaussian noise stimulus set. TWM is funded by The Wellcome Trust (306384/Z/23/Z; 318818/Z/24/Z) and Gatsby Charitable Foundation (GAT4057), JB is funded by EMBO (ALTF 415–2024), and CC is funded by The Wellcome Trust (200790/Z/16/Z), The Simons Foundation (564408), EPSRC (EP/R035806/1), and the ERC (MotorAdapt 101169605).

## Additional information

### Funding

| Funder | Grant reference number | Author |
| --- | --- | --- |
| Wellcome Trust | 318818/Z/24/Z | Troy W Margrie |
| Wellcome Trust | 10.35802/306384 | Troy W Margrie |
| Gatsby Charitable Foundation | GAT4057 | Troy W Margrie |
| European Molecular Biology Organization | ALTF 415-2024 | Joel Bauer |
| Wellcome Trust | 10.35802/200790 | Claudia Clopath |
| Simons Foundation | 564408 | Claudia Clopath |

| Funder | Grant reference number | Author |
|---|---|---|
| Engineering and Physical Sciences Research Council | EP/R035806/1 | Claudia Clopath |
| European Research Council | MotorAdapt 101169605 | Claudia Clopath |

The funders had no role in study design, data collection and interpretation, or the decision to submit the work for publication. For the purpose of Open Access, the authors have applied a CC BY public copyright license to any Author Accepted Manuscript version arising from this submission.

### Author contributions

Joel Bauer, Conceptualization, Software, Formal analysis, Funding acquisition, Validation, Investigation, Visualization, Methodology, Writing – original draft, Writing – review and editing; Troy W Margrie, Claudia Clopath, Conceptualization, Resources, Supervision, Funding acquisition, Project administration, Writing – review and editing

### Author ORCIDs

Joel Bauer ⓘ https://orcid.org/0000-0001-5858-166X
Troy W Margrie ⓘ https://orcid.org/0000-0002-5526-4578
Claudia Clopath ⓘ https://orcid.org/0000-0003-4507-8648

Reviewer #2 (Public review): https://doi.org/10.7554/eLife.105081.3.sa1
Reviewer #3 (Public review): https://doi.org/10.7554/eLife.105081.3.sa2
Author response https://doi.org/10.7554/eLife.105081.3.sa3

## Additional files

### Supplementary files

MDAR checklist

### Data availability

The code is available at https://github.com/Joel-Bauer/movie_reconstruction_code (copy archived at *Bauer, 2025*).

The following previously published datasets were used:

| Author(s) | Year | Dataset title | Dataset URL | Database and Identifier |
|---|---|---|---|---|
| Fahey P, Turishcheva P, Hansel L, Froebe R, Ponder K, Vystrcilová M, Qiu Y, Willeke K, Bashiri M, Tolias A, Sinz A, Ecker A | 2023 | The Dynamic Sensorium competition for predicting large-scale mouse visual cortex activity from videos - Dataset | https://gin.g-node.org/pollytur/Sensorium2023Data | G-Node Gin, Sensorium2023Data |
| Fahey P, Turishcheva P, Hansel L, Froebe R, Ponder K, Vystrcilová M, Qiu Y, Willeke K, Bashiri M, Tolias A, Sinz A, Ecker A | 2023 | The Dynamic Sensorium competition for predicting large-scale mouse visual cortex activity from videos - Dataset | https://gin.g-node.org/pollytur/sensorium_2023_data | G-Node Gin, sensorium_2023_data |

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

# Appendix 1

## Supplemental material

**Appendix 1 – algorithm 1. Movie reconstruction.**

1: Parameters:

2: Dynamic Neural Encoding Model (DNEM), Ground truth neuronal response $y$, predicted neuronal responses $\hat{y}$, predicted input video $\hat{x}$, video width $w$=64, video height $h$=36, number of frames $n$=300, transparency mask α, sliding window size $k$=32, sliding window stride $s$=8, total number of windows N←[2+$(n-k)/s$], learning rate $lr = 1000$, $\beta_1 = 0.9$, loss function $Loss(y, \hat{y}) = \sum \hat{y}_{neuron} - y_{neuron} \log(\hat{y}_{neuron} + 10^{-8})$, number of epochs $epochs = 1000$, number of model instances q=7

3: **Objective:** $F(y, \hat{x}) = Loss(y, DNEM(\hat{x}))$

4: Initialize variables:

5: $\hat{x}_{f,h,w} \leftarrow 125$

6: $G_{i,f,h,w} \leftarrow 0$

7: $g_{f,h,w} \leftarrow 0$

8: $m_{f,h,w} \leftarrow 0$

9: **for** iteration $t = 1, \ldots, epochs$ **do**

10: Gradients across all windows:

11: **for** iteration $i = 1, \ldots, N$ **do**

12: **if** $i < N$ **then**

13: $f \leftarrow [s(i-1), \ldots, k + s(i-1)]$

14: **else if** $i = N$ **then**

15: $f \leftarrow [n - k, \ldots, n]$

16: **end if**

17: $G_{i,f} \leftarrow \nabla F(y_f, \hat{x}_f)$

18: $G_{i,f} \leftarrow \frac{G_{i,f}}{\|G_{i,f}\| + 10^{-6}}$

19: **end for**

20: Average gradients across windows: $g \leftarrow \frac{1}{N} \sum_{i=1}^{N} G_i$

21: Clip and mask gradients:

22: $g \leftarrow \text{clip}(g, -1, 1)$

23: $g_{f,h,w} \leftarrow g_{f,h,w} \odot (\alpha_{h,w} > 0.5)$

24: Update movie:

25: **if** $t < 10$ **then**

26: $lr_{current} \leftarrow lr \frac{t}{10}$

27: **else if** $t > 10$ **then**

28: $lr_{current} \leftarrow lr$

29: **end if**

30: $lr_{current} \leftarrow lr_{current} \sqrt{1 - 0.999^t}/(1 - \beta_1^t)$

31: $m \leftarrow \beta_1 \cdot m + (1 - \beta_1) \cdot g$

32: $\hat{m} \leftarrow \frac{m}{1 - \beta_1^t}$

33: $\hat{x} \leftarrow \hat{x} - lr_{current} \cdot \hat{m}$

34: Clip movie: $\hat{x} \leftarrow \text{clip}(\hat{x}, 0, 255)$

35: **end for**

36: **Ensembling:** $\hat{x} \leftarrow \frac{1}{q} \sum_{i=1}^{q} \hat{x}$

37: **Denoise (optional):** $\hat{x} \leftarrow GaussianBlur3D(\hat{x}, \sigma = (0.5, 0.5, 0.5))$

38: **Mask movie:** $\hat{x}_{f,h,w} \leftarrow \hat{x}_{f,h,w} \odot (\alpha_{h,w} > 1)$

Appendix 1 – algorithm 2. Mask training.

1: Parameters:
2: Dynamic Neural Encoding Model (DNEM), Ground truth neuronal response $y$, predicted neuronal responses $\hat{y}$, ground
   truth input video $x$, background video *background*, video width $w$=64, video height $h$=36, frames $f$, number of frames $n$=32,
   transparency mask $\alpha$, learning rate $lr = 10$, loss function $Loss(y, \hat{y}, \alpha) = MSE(y, \hat{y}(1 - \bar{\alpha}))$,
   number of epochs $epochs = 1000$, alpha blending function $Blend(x, background, \alpha) = x \odot \alpha + background \odot (1 - \alpha)$

3: Objective:
4: $F(y, x, background, \alpha) = Loss(y, DNEM(Blend(x, background, \text{clip}(\alpha, -1, 1))), \alpha)$

5: Initialize variables:
6: $\alpha_{h,w} \leftarrow U(0, 0.5)$

7: **for** iteration $t = 1, \ldots, epochs$ **do**

8:      $x \leftarrow$ get random video
9:      $background \leftarrow$ get different random video
10:     $background \leftarrow Flip(background, dim = 1)$
11:     $background \leftarrow Flip(background, dim = 2)$
12:     $background \leftarrow Flip(background, dim = 3)$

13:     Gradients with respect to transparency mask:
14:     $G \leftarrow \frac{\partial F(y, x, \text{background}, \alpha)}{\partial \alpha}$
15:     $G \leftarrow \frac{G}{\|G\| + 10^{-6}}$

16:     Average gradients across frames:
17:     $g \leftarrow \frac{1}{n} \sum_{f=1}^{n} G_f$

18: Clip and mask gradients:
19:     $g \leftarrow \text{clip}(g, -1, 1)$
20:     $g \leftarrow GaussianBlur2D(g, \sigma = (5, 5))$

21:     Update mask:
22:     $\alpha \leftarrow \alpha - lr \cdot g$

23: **end for**

