## [Editor Report · eLife Assessment]

This **valuable** study uses state-of-the-art neural encoding and video reconstruction methods to achieve a substantial improvement in video reconstruction quality from mouse neural data. It provides a **convincing** demonstration of how reconstruction performance can be improved by combining these methods. The goal of the study was improving reconstruction performance rather than advancing theoretical understanding of neural processing, so the results will be of practical interest to the brain decoding community.

---

## [Referee Report · Reviewer #2 (Public review)]

Summary:

This is an interesting study exploring methods for reconstructing visual stimuli from neural activity in the mouse visual cortex. Specifically, it uses a competition dataset (published in the Dynamic Sensorium benchmark study) and a recent winning model architecture (DNEM, dynamic neural encoding model) to recover visual information stored in ensembles of mouse visual cortex.

Strengths:

This is a great start for a project addressing visual reconstruction. It is based on physiological data obtained at a single-cell resolution, the stimulus movies were reasonably naturalistic and representative of the real world, the study did not ignore important correlates such as eye position and pupil diameter, and of course, the reconstruction quality exceeded anything achieved by previous studies. There appear to be no major technical flaws in the study, and some potential confounds were addressed upon revision. The study is an enjoyable read.

Weaknesses:

The study is technically competent and benchmark-focused, but without significant conceptual or theoretical advances. The inclusion of neuronal data broadens the study's appeal, but the work does not explore potential principles of neural coding, which limits its relevance for neuroscience and may create some disappointment to some neuroscientists. The authors are transparent that their goal was methodological rather than explanatory, but this raises the question of why neuronal data were necessary at all, as more significant reconstruction improvements might be achievable using noise-less artificial video encoders alone (network-to-network decoding approaches have been done well by teams such as Han, Poggio, and Cheung, 2023, ICML). Yet, even within the methodological domain, the study does not articulate clear principles or heuristics that could guide future progress. The finding that more neurons improve reconstruction aligns with well-established results in the literature that show that higher neuronal numbers improve decoding in general (for example, Hung, Kreiman, Poggio, and DiCarlo, 2005) and thus may not constitute a novel insight.

Specific issues:

(1) The study showed that it could achieve high-quality video reconstructions from mouse visual cortex activity using a neural encoding model (DNEM), recovering 10-second video sequences and approaching a two-fold improvement in pixel-by-pixel correlation over attempts. As a reader, I was left with the question: okay, does this mean that we should all switch to DNEM for our investigations of mouse visual cortex? What makes this encoding model special? It is introduced as "a winning model of the Sensorium 2023 competition which achieved a score of 0.301...single trial correlation between predicted and ground truth neuronal activity," but as someone who does not follow this competition (most eLife readers are not likely to do so, either), I do not know how to gauge my response. Is this impressive? What is the best theoretical score, given noise and other limitations? Is the model inspired by the mouse brain in terms of mechanisms or architecture, or was it optimized to win the competition by overfitting it to the nuances of the data set? Of course, I know that as a reader, I am invited to read the references, but the study would stand better on its own, if it clarified how its findings depended on this model.

The revision helpfully added context to the Methods about the range of scores achieved by other models, but this information remains absent from the Abstract and other important sections. For instance, the Abstract states, "We achieve a pixel-level correlation of 0.57 between the ground truth movie and the reconstructions from single-trial neural responses," yet this point estimate (presented without confidence intervals or comparisons to controls) lacks meaning for readers who are not told how it compares to prior work or what level of performance would be considered strong. Without such context, the manuscript undercuts potentially meaningful achievements.

(2) Along those lines, the authors conclude that "the number of neurons in the dataset and the use of model ensembling are critical for high-quality reconstructions." If true, these principles should generalize across network architectures. I wondered whether the same dependencies would hold for other network types, as this could reveal more general insights. The authors replied that such extensions are expected (since prior work has shown similar effects for static images) but argued that testing this explicitly would require "substantial additional work," be "impractical," and likely not produce "surprising results." While practical difficulty alone is not a sufficient reason to leave an idea untested, I agree that the idea that "more neurons would help" would be unsurprising. The question then becomes: given that this is a conclusion already in the field, what new principle or understanding has been gained in this study?

(3) One major claim was that the quality of the reconstructions depended on the number of neurons in the dataset. There were approximately 8000 neurons recorded per mouse. The correlation difference between the reconstruction achieved by 1000 neurons and 8000 neurons was ~0.2. Is that a lot or a little? One might hypothesize that 7000 additional neurons could contribute more information, but perhaps, those neurons were redundant if their receptive fields are too close together or if they had the same orientation or spatiotemporal tuning. How correlated were these neurons in response to a given movie? Why did so many neurons offer such a limited increase in correlation? Originally, this question was meant to prompt deeper analysis of the neural data, but the authors did not engage with it, suggesting a limited understanding of the neuronal aspects of the dataset.

(4) We appreciated the experiments testing the capacity of the reconstruction process, by using synthetic stimuli created under a Gaussian process in a noise-free way. But this originally further raised questions: what is the theoretical capability for reconstruction of this processing pipeline, as a whole? Is 0.563 the best that one could achieve given the noisiness and/or neuron count of the Sensorium project? What if the team applied the pipeline to reconstruct the activity of a given artificial neural network's layer (e.g., some ResNet convolutional layer), using hidden units as proxies for neuronal calcium activity? In the revision, this concern was addressed nicely in the review in Supplementary Figure 3C. Also, one appreciates that as a follow up, the team produced error maps (New Figure 6) that highlight where in the frames the reconstruction are likely to fail. But the maps went unanalyzed further, and I am not sure if there was a systematic trend in the errors.

(5) I was encouraged by Figure 4, which shows how the reconstructions succeeded or failed across different spatial frequencies. The authors note that "the reconstruction process failed at high spatial frequencies," yet it also appears to struggle with low spatial frequencies, as the reconstructed images did not produce smooth surfaces (e.g., see the top rows of Figures 4A and 4B). In regions where one would expect a single continuous gradient, the reconstructions instead display specular, high-frequency noise. This issue is difficult to overlook and might deserve further discussion.

---

## [Referee Report · Reviewer #3 (Public review)]

Summary:

This paper presents a method for reconstructing input videos shown to a mouse from the simultaneously recorded visual cortex activity (two-photon calcium imaging data). The publicly available experimental dataset is taken from a recent brain-encoding challenge, and the (publicly available) neural network model that serves to reconstruct the videos is the winning model from that challenge (by distinct authors). The present study applies gradient-based input optimization by backpropagating the brain-encoding error through this selected model (a method that has been proposed in the past, with other datasets). The main contribution of the paper is, therefore, the choice of applying this existing method to this specific dataset with this specific neural network model. The quantitative results appear to go beyond previous attempts at video input reconstruction (although measured with distinct datasets). The conclusions have potential practical interest for the field of brain decoding, and theoretical interest for possible future uses in functional brain exploration.

Strengths:

The authors use a validated optimization method on a recent large-scale dataset, with a state-of-the-art brain encoding model. The use of an ensemble of 7 distinct model instances (trained on distinct subsets of the dataset, with distinct random initializations) significantly improves the reconstructions. The exploration of the relation between reconstruction quality and number of recorded neurons will be useful to those planning future experiments.

Weaknesses:

The main contribution is methodological, and the methodology combines pre-existing components without any new original component.

---

## [Author Response]

The following is the authors’ response to the current reviews.

**Public Reviews:**

**Reviewer #2 (Public review):**
Summary:This is an interesting study exploring methods for reconstructing visual stimuli from neural activity in the mouse visual cortex. Specifically, it uses a competition dataset (published in the Dynamic Sensorium benchmark study) and a recent winning model architecture (DNEM, dynamic neural encoding model) to recover visual information stored in ensembles of mouse visual cortex.Strengths:This is a great start for a project addressing visual reconstruction. It is based on physiological data obtained at a single-cell resolution, the stimulus movies were reasonably naturalistic and representative of the real world, the study did not ignore important correlates such as eye position and pupil diameter, and of course, the reconstruction quality exceeded anything achieved by previous studies. There appear to be no major technical flaws in the study, and some potential confounds were addressed upon revision. The study is an enjoyable read.Weaknesses:The study is technically competent and benchmark-focused, but without significant conceptual or theoretical advances. The inclusion of neuronal data broadens the study's appeal, but the work does not explore potential principles of neural coding, which limits its relevance for neuroscience and may create some disappointment to some neuroscientists. The authors are transparent that their goal was methodological rather than explanatory, but this raises the question of why neuronal data were necessary at all, as more significant reconstruction improvements might be achievable using noise-less artificial video encoders alone (network-to-network decoding approaches have been done well by teams such as Han, Poggio, and Cheung, 2023, ICML). Yet, even within the methodological domain, the study does not articulate clear principles or heuristics that could guide future progress. The finding that more neurons improve reconstruction aligns with well-established results in the literature that show that higher neuronal numbers improve decoding in general (for example, Hung, Kreiman, Poggio, and DiCarlo, 2005) and thus may not constitute a novel insight.

We thank the reviewer for this second round of comments and hope we were able to address the remaining points below.

Indeed, using surrogate noiseless data is interesting and useful when developing such methods, or to demonstrate that they work in principle. But in order to evaluate if they really work in practice, we need to use real neuronal data. While we did not try movie reconstruction from layers within artificial neural networks as surrogate data, in Supplementary Figure 3C we provide the performance of our method using simulated/predicted neuronal responses from the dynamic neural encoding model alongside real neuronal responses.

Specific issues:(1)The study showed that it could achieve high-quality video reconstructions from mouse visual cortex activity using a neural encoding model (DNEM), recovering 10-second video sequences and approaching a two-fold improvement in pixel-by-pixel correlation over attempts. As a reader, I was left with the question: okay, does this mean that we should all switch to DNEM for our investigations of mouse visual cortex? What makes this encoding model special? It is introduced as "a winning model of the Sensorium 2023 competition which achieved a score of 0.301...single trial correlation between predicted and ground truth neuronal activity," but as someone who does not follow this competition (most eLife readers are not likely to do so, either), I do not know how to gauge my response. Is this impressive? What is the best theoretical score, given noise and other limitations? Is the model inspired by the mouse brain in terms of mechanisms or architecture, or was it optimized to win the competition by overfitting it to the nuances of the data set? Of course, I know that as a reader, I am invited to read the references, but the study would stand better on its own, if it clarified how its findings depended on this model.The revision helpfully added context to the Methods about the range of scores achieved by other models, but this information remains absent from the Abstract and other important sections. For instance, the Abstract states, "We achieve a pixel-level correlation of 0.57 between the ground truth movie and the reconstructions from single-trial neural responses," yet this point estimate (presented without confidence intervals or comparisons to controls) lacks meaning for readers who are not told how it compares to prior work or what level of performance would be considered strong. Without such context, the manuscript undercuts potentially meaningful achievements.

We appreciate that the additional information about the performance of the SOTA DNEM to predict neural responses could be made more visible in the paper and will therefore move it from the methods to the results section instead:

Line 348 “This model achieved an average single-trial correlation between predicted and ground truth neural activity of 0.291 during the competition, this was later improved to 0.301. The competition benchmark models achieved 0.106, 0.164 and 0.197 single-trial correlation, while the third and second place models achieved 0.243 and 0.265. Across the models, a variety of architectural components were used, including 2D and 3D convolutional layers, recurrent layers, and transformers, to name just a few.” will be moved to the results.

With regard to the lack of context for the performance of our reconstruction in the abstract, we may have overcorrected in the previous revision round and have tried to find a compromise which gives more context to the pixel-level correlation value:

Abstract: “We achieve a pixel-level correlation of 0.57 (95% CI [0.54, 0.60]) between ground-truth movies and single-trial reconstructions. Previous reconstructions based on awake mouse V1 neuronal responses to static images achieved a pixel-level correlation of 0.238 over a similar retinotopic area.”

(2) Along those lines, the authors conclude that "the number of neurons in the dataset and the use of model ensembling are critical for high-quality reconstructions." If true, these principles should generalize across network architectures. I wondered whether the same dependencies would hold for other network types, as this could reveal more general insights. The authors replied that such extensions are expected (since prior work has shown similar effects for static images) but argued that testing this explicitly would require "substantial additional work," be "impractical," and likely not produce "surprising results." While practical difficulty alone is not a sufficient reason to leave an idea untested, I agree that the idea that "more neurons would help" would be unsurprising. The question then becomes: given that this is a conclusion already in the field, what new principle or understanding has been gained in this study?

As mentioned in our previous round of revisions, we chose not to pursue the comparison of reconstructions using different model architectures in this manuscript because we did not think it would add significant insights to the paper given the amount of work it would require, and we are glad the reviewer agrees.

While the fact that more neurons result in better reconstructions is unsurprising, how quickly performance drops off will depend on the robustness of the method, and on the dimensionality of the decoding/reconstruction task (decoding grating orientation likely requires fewer neurons than gray scale image reconstruction, which in turn likely requires fewer neurons than full color movie reconstruction). How dependent input optimization based image/movie reconstruction is on population size has not been shown, so we felt it was useful for readers to know how well movie reconstruction works with our method when recording from smaller numbers of neurons.

(3) One major claim was that the quality of the reconstructions depended on the number of neurons in the dataset. There were approximately 8000 neurons recorded per mouse. The correlation difference between the reconstruction achieved by 1000 neurons and 8000 neurons was ~0.2. Is that a lot or a little? One might hypothesize that 7000 additional neurons could contribute more information, but perhaps, those neurons were redundant if their receptive fields are too close together or if they had the same orientation or spatiotemporal tuning. How correlated were these neurons in response to a given movie? Why did so many neurons offer such a limited increase in correlation? Originally, this question was meant to prompt deeper analysis of the neural data, but the authors did not engage with it, suggesting a limited understanding of the neuronal aspects of the dataset.

We apologize that we did not engage with this comment enough in the previous round. We assumed that the question arose because there was a misunderstanding about figure 5: 1000 not 1 neuron is sufficient to reconstruct the movies to a pixel-level correlation of 0.344. Of course, the fact that increasing the number of neurons from 1000 to 8000 only increased the reconstruction performance from 0.344 to 0.569 (65% increase in correlation) is still worth discussing. To illustrate this drop in performance qualitatively, we show 3 example frames from movie reconstructions using 1000-8000 neurons in Author response image 1.

**Author response image 1. sa3fig1:** 3 example frames from reconstructions using different numbers of neurons.

As the reviewer points out, the diminishing returns of additional neurons to reconstruction performance is at least partly because there is redundancy in how a population of neurons represents visual stimuli. In supplementary figure S2, we inferred the on-off receptive fields of the neurons and show that visual space is oversampled in terms of the receptive field positions in panel C. However, the exact slope/shape of the performance vs population size curve we show in Figure 5 will also depend on the maximum performance of our reconstruction method, which is limited in spatial resolution (Figure 4 & Supplementary Figure S5). It is possible that future reconstruction approaches will require fewer neurons than ours, so we interpret this curve rather as a description of the reconstruction method itself than a feature of the underlying neuronal code. For that reason, we chose caution and refrained from making any claims about neuronal coding principles based on this plot.

(4) We appreciated the experiments testing the capacity of the reconstruction process, by using synthetic stimuli created under a Gaussian process in a noise-free way. But this originally further raised questions: what is the theoretical capability for reconstruction of this processing pipeline, as a whole? Is 0.563 the best that one could achieve given the noisiness and/or neuron count of the Sensorium project? What if the team applied the pipeline to reconstruct the activity of a given artificial neural network's layer (e.g., some ResNet convolutional layer), using hidden units as proxies for neuronal calcium activity? In the revision, this concern was addressed nicely in the review in Supplementary Figure 3C. Also, one appreciates that as a follow up, the team produced error maps (New Figure 6) that highlight where in the frames the reconstruction are likely to fail. But the maps went unanalyzed further, and I am not sure if there was a systematic trend in the errors.

We are happy to hear that we were able to answer the reviewers’ question of what the maximum theoretical performance of our reconstruction process is in figure 3C. Regarding systematic trends in the error maps, we also did not observe any clear systematic trends. If anything, we noticed that some moving edges were shifted, but we do not think we can quantify this effect with this particular dataset.

(5) I was encouraged by Figure 4, which shows how the reconstructions succeeded or failed across different spatial frequencies. The authors note that "the reconstruction process failed at high spatial frequencies," yet it also appears to struggle with low spatial frequencies, as the reconstructed images did not produce smooth surfaces (e.g., see the top rows of Figures 4A and 4B). In regions where one would expect a single continuous gradient, the reconstructions instead display specular, high-frequency noise. This issue is difficult to overlook and might deserve further discussion.

Thank you for pointing this out, this is indeed true. The reconstructions do have high frequency noise. We mention this briefly in line 102 “Finally, we applied a 3D Gaussian filter with sigma 0.5 pixels to remove the remaining static noise (Figure S3) and applied the evaluation mask.” In revisiting this sentence, we think it is more appropriate to replace “remove” with “reduce”. This noise is more visible in the Gaussian noise stimuli (Figure 4) because we did not apply the 3D Gaussian filter to these reconstructions, in case it interfered with the estimates of the reconstruction resolution limits.

Given that the Gaussian noise and drifting grating stimuli reconstructions were from predicted activity (“noise-free”), this high-frequency noise is not biological in origin and must therefore come from errors in our reconstruction process. This kind of high-frequency noise has previously been observed in feature visualization (optimizing input to maximize the activity of a specific node within a neural network to visualize what that node encodes; Olah, et al., "Feature Visualization", https://distill.pub/2017/feature-visualization/, 2017). It is caused by a kind of overfitting, whereby a solution to the optimization is found that is not “realistic”. Ways of combating this kind of noise include gradient smoothing, image smoothing, and image transformations during optimization, but these methods can restrict the resolution of the features that are recovered. Since we were more interested in determining the maximum resolution of stimuli that can be reconstructed in Figure 4 and Supplementary Figures 5-6, we chose not to apply these methods.

**Reviewer #3 (Public review):**
Summary:This paper presents a method for reconstructing input videos shown to a mouse from the simultaneously recorded visual cortex activity (two-photon calcium imaging data). The publicly available experimental dataset is taken from a recent brain-encoding challenge, and the (publicly available) neural network model that serves to reconstruct the videos is the winning model from that challenge (by distinct authors). The present study applies gradient-based input optimization by backpropagating the brain-encoding error through this selected model (a method that has been proposed in the past, with other datasets). The main contribution of the paper is, therefore, the choice of applying this existing method to this specific dataset with this specific neural network model. The quantitative results appear to go beyond previous attempts at video input reconstruction (although measured with distinct datasets). The conclusions have potential practical interest for the field of brain decoding, and theoretical interest for possible future uses in functional brain exploration.Strengths:The authors use a validated optimization method on a recent large-scale dataset, with a state-of-the-art brain encoding model. The use of an ensemble of 7 distinct model instances (trained on distinct subsets of the dataset, with distinct random initializations) significantly improves the reconstructions. The exploration of the relation between reconstruction quality and number of recorded neurons will be useful to those planning future experiments.Weaknesses:The main contribution is methodological, and the methodology combines pre-existing components without any new original component.

We thank the reviewer for their balanced assessment of our manuscript.

The following is the authors’ response to the original reviews.

**Public Reviews:**

**Reviewer #1 (Public review):**
Summary:This paper presents a method for reconstructing videos from mouse visual cortex neuronal activity using a state-of-the-art dynamic neural encoding model. The authors achieve high-quality reconstructions of 10-second movies at 30 Hz from two-photon calcium imaging data, reporting a 2-fold increase in pixel-by-pixel correlation compared to previous methods. They identify key factors for successful reconstruction including the number of recorded neurons and model ensembling techniques.Strengths:(1) A comprehensive technical approach combining state-of-the-art neural encoding models with gradient-based optimization for video reconstruction.(2) Thorough evaluation of reconstruction quality across different spatial and temporal frequencies using both natural videos and synthetic stimuli.(3) Detailed analysis of factors affecting reconstruction quality, including population size and model ensembling effects.(4) Clear methodology presentation with well-documented algorithms and reproducible code.(5) Potential applications for investigating visual processing phenomena like predictive coding and perceptual learning.

We thank the reviewer for taking the time to provide this valuable feedback. We would like to add that in our eyes one additional main contribution is the step of going from reconstruction of static images to dynamic videos. We trust that in the revised manuscript, we have now made the point more explicit that static image reconstruction relies on temporally averaged responses, which negates the necessity of having to account for temporal dynamics altogether.

Weaknesses:The main metric of success (pixel correlation) may not be the most meaningful measure of reconstruction quality:High correlation may not capture perceptually relevant features.Different stimuli producing similar neural responses could have low pixel correlations The paper doesn't fully justify why high pixel correlation is a valuable goal

This is a very relevant point. In retrospect, perhaps we did not justify this enough. Sensory reconstruction typically aims to reconstruct sensory input based on brain activity as faithfully as possible. A brain-to-image decoder might therefore be trained to produce images as close to the original input as possible. The loss function to train the decoder would therefore be image similarity on the pixel level. In that case, evaluating reconstruction performance based on pixel correlation is somewhat circular.

However, when reconstructing videos, we optimize the input video in terms of its perceptual similarity to the original video and only then evaluate pixel-level similarity. The perceptual similarity metric we optimize for is the estimate of how the neurons in mouse V1 respond to that video. We then evaluate the similarity of this perceptually optimized video to the original input video with pixel-level correlation. In other words, we optimize for perceptual similarity and then evaluate pixel similarity. If our method optimized pixel-level similarity, then we would agree that perceptual similarity is a more relevant evaluation metric. We do not think it was clear in our original submission that our optimization loss function is a perceptual loss function, and have now made this clearer in Figure 1C-D and have clarified this in the results section, line 70:

“In effect, we optimized the input video to be perceptually similar with respect to the recorded neurons.”

And in line 110:

“Because our optimization of the movies was based on a perceptual loss function, we were interested in how closely these movies matched the originals on the pixel level.”

We chose to use pixel correlation to measure pixel-level similarity for several reasons. (1) It has been used in the past to evaluate reconstruction performance (Yoshida et al., 2020), (2) It is contrast and luminance insensitive, (3) correlation is a common metric so most readers will have an intuitive understanding of how it relates to the data.

To further highlight why pixel similarity might be interesting to visualize, we have included additional analysis in Figure 6 illustrating pixel-level differences between reconstructions from experimentally recorded activity and predicted activity.

We expect that the type of perceptual similarity the reviewer is alluding to is pretrained neural network image embedding similarity (Zhang et al., 2018: https://doi.org/10.48550/arXiv.1801.03924). While these metrics seem to match human perceptual similarity, it is unclear if they reflect mouse vision. We did try to compare the embedding similarity from pretrained networks such as VGG16, but got results suggesting the reconstructed frames were no more similar to the ground truth than random frames, which is obviously not true. This might be because the ground truth videos were too different in resolution from the training data of these networks and because these metrics are typically very sensitive to decreases in resolution.

The best alternative approach to evaluate mouse perceptual similarity would be to show the reconstructed videos to the same animals while recording the same neurons and to compare these neural activation patterns to those evoked by the original ground truth videos. This has been done for static images in the past: [11], found that static image reconstructions generated using gradient descent evoked more similar trial-averaged (40 trials) responses to those evoked by ground truth images compared to other reconstruction methods. Unfortunately, we are currently not able to perform these in vivo experiments, which is why we used publicly available data for the current paper. We plan to use this method in the future. But this method is also not flawless as it assumes that the average response to an image is the best reflection of how that image is represented, which may not be the case for an individual trial.

As far as we are aware, there is currently no method that, given a particular activity pattern in response to an image/video, can produce an image/video that induces a neural activity pattern that is closer to the original neural response than simply showing the same image/video again. Hypothetically, such a stimulus exists because of various visual processing phenomena we mention in our discussion (e.g., predictive coding and selective attention), which suggest that the image that is represented by a population of neurons likely differs from the original sensory input. In other words, what the brain represents is an interpretation of reality not a pure reflection. Experimentally verifying this is difficult, as these variations might be present on a single trial level. The first step towards establishing a method that captures the visual representation of a population of neurons is sensory reconstruction, where the aim is to get as close as possible to the original sensory input. We think pixel-level correlation is a stringent and interpretable metric for this purpose, particularly when optimizing for perceptual similarity rather than image similarity directly.

Comparison to previous work (Yoshida et al.) has methodological concerns: Direct comparison of correlation values across different datasets may be misleading; Large differences in the number of recorded neurons (10x more in the current study); Different stimulus types (dynamic vs static) make comparison difficult; No implementation of previous methods on the current dataset or vice versa.

Yes, we absolutely agree that direct comparison to previous static image reconstruction methods is problematic. We primarily do so because we think it is standard practice to give related baselines. We agree that direct comparison of the performance of video reconstruction methods to image reconstruction methods is not really possible. It does not make sense to train and apply a dynamic model on a static image data set where neural activity is time-averaged, as the temporal kernels could not be learned. Conversely, for a static model, which expects a single image as input and predicts time averaged responses, it does not make sense to feed it a series of temporally correlated movie frames and to simply concatenate the resulting activity perdition. The static model would need to be substantially augmented to incorporate temporal dynamics, which in turn would make it a new method. This puts us in the awkward position of being expected to compare our video reconstruction performance to previous image reconstruction methods without a fair way of doing so. We have now added these caveats in line 119:

“However, we would like to stress that directly comparing static image reconstruction methods with movie reconstruction approaches is fundamentally problematic, as they rely on different data types both during training and evaluation (temporally averaged vs continuous neural activity, images flashed at fixed intervals vs continuous movies).”

We have also toned down the language, emphasising the comparison to previous image reconstruction performance in the abstract, results, and conclusion.

Abstract: We removed “We achieve a ~2-fold increase in pixel-by-pixel correlation compared to previous state-of-the-art reconstructions of static images from mouse V1, while also capturing temporal dynamics.” and replaced with “We achieve a pixel-level correction of 0.57 between the ground truth movie and the reconstructions from single-trial neural responses.”

Discussion: we removed “In conclusion, we reconstruct videos presented to mice based on the activity of neurons in the mouse visual cortex, with a ~2-fold improvement in pixel-by-pixel correlation compared to previous static image reconstruction methods.” and replaced with “In conclusion, we reconstruct videos presented to mice based on single-trial activity of neurons in the mouse visual cortex.”

We have also removed the performance table and have instead added supplementary figure 3 with in-depth comparison across different versions of our reconstruction method (variations of masking, ensembling, contrast & luminance matching, and Gaussian blurring).

Limited exploration of how the reconstruction method could provide insights into neural coding principles beyond demonstrating technical capability.

The aim of this paper was not to reveal principles of neural coding. Instead, we aimed to achieve the best possible performance of video reconstructions and to quantify the limitations. But to highlight its potential we have added two examples of how sensory reconstruction has been applied in human vision research in line 321:

“Although fMRI-based reconstruction techniques are starting to be used to investigate visual phenomena in humans (such as illusions [Cheng et al., 2023] and mental imagery [Shen et al., 2019; Koide-Majima et al., 2024; Kalantari et al., 2025]), visual processing phenomena are likely difficult to investigate using existing fMRI-based reconstruction approaches, due to the low spatial and temporal resolution of the data.”

We have also added a demonstration of how this method could be used to investigate which parts of a reconstruction from a single trial response differs from the model's prediction (Figure 6). We do this by calculating pixel-level differences between reconstructions from the recorded neural activity and reconstructions from the expected neural activity (predicted activity by the neural encoding model). Although difficult to interpret, this pixel-by-pixel error map could represent trial-by-trial deviations of the neural code from pure sensory representation. But at this point we cannot know whether these errors are nothing more than errors in the reconstruction process. To derive meaningful interpretations of these maps would require a substantial amount of additional work and in vivo experiments and so is outside the scope of this paper, but we include this additional analysis now to highlight (a) why pixel-level similarity might be interesting to quantify and visualize and (b) to demonstrate how video reconstruction could be used to provide insights into neural coding, namely as a tool to identify how sensory representations differ from a pure reflection of the visual input.

The claim that "stimulus reconstruction promises a more generalizable approach" (line 180) is not well supported with concrete examples or evidence.

What we mean by generalizable is the ability to apply reconstruction to novel stimuli, which is not possible for stimulus classification. We now explain this better in the paragraph in line 211:

“Stimulus identification, i.e. identifying the most likely stimulus from a constrained set, has been a popular approach for quantifying whether a population of neurons encodes the identity of a particular stimulus [Földiák, 1993, Kay et al., 2008]. This approach has, for instance, been used to decode frame identity within a movie [Deitch et al., 2021, Xia et al., 2021, Schneider et al., 2023, Chen et al.,2024]. Some of these approaches have also been used to reorder the frames of the ground truth movie [Schneider et al., 2023] based on the decoded frame identity. Importantly, stimulus identification methods are distinct from stimulus reconstruction where the aim is to recreate what the sensory content of a neuronal code is in a way that generalizes to new sensory stimuli [Rakhimberdina et al., 2021]. This is inherently a more demanding task because the range of possible solutions is much larger. Although stimulus identification is a valuable tool for understanding the information content of a population code, stimulus reconstruction could provide a more generalizable approach, because it can be applied to novel stimuli.”

All the stimuli we reconstructed were not in the training set of the model, i.e., novel. We have also downed down the claim: we have replaced “promises” with “could provide”.

The paper would benefit from addressing how the method handles cases where different stimuli produce similar neural responses, particularly for high-speed moving stimuli where phase differences might be lost in calcium imaging temporal resolution.

Thank you for this suggestion, we think this is a great question. Calcium dynamics are slow and some of the high temporal frequency information could indeed be lost, particularly phase information. In other words, when the stimulus has high temporal frequency information, it is harder to decode spatial information because of the slow calcium dynamics. Ideally, we would look at this effect using the drifting grating stimuli; however, this is problematic because we rely on predicted activity from the SOTA DNEM, and due to the dilation of the first convolution, the periodic grating stimulus causes aliasing. At 15Hz, when the temporal frequency of the stimulus is half the movie frame rate, the model is actually being given two static images, and so the predicted activity is the interleaved activity evoked by two static images. We therefore do not think using the grating stimuli is a good idea. But we have used the Gaussian stimuli as it is not periodic, and is therefore less of a problem.

We have now also reconstructed phase-inverted Gaussian noise stimuli and plotted the video correlation between the reconstructions from activity evoked by phase-inverted stimuli. On the one hand, we find that even for the fastest changing stimuli, the correlation between the reconstructions from phase inverted stimuli are negative, meaning phase information is not lost at high temporal frequencies. On the other hand, for the highest spatial frequency stimuli, the correlation is negative. So, the predicted neural activity (and therefore the reconstructions) are phase-insensitive when the spatial frequency is higher than the reconstruction resolution limit we identified (spatial length constant of 1 pixel, or 3.38 degrees). Beyond this limit, the DNEM predicts activity in response to phase-inverted stimuli, which, when used for reconstruction, results in movies which are more similar to each other than the stimulus that actually evokes them.

However, not all information is lost at these high spatial frequencies. If we plot the Shannon entropy in the spatial domain or the motion energy in the temporal domain, we find that even when the reconstructions fail to capture the stimulus at a pixel-specific level (spatial length constant of 1 pixel, or 3.38 degrees), they do capture the general spatial and temporal qualities of the videos.

We have added these additional analyses to Figure 4 and Supplementary Figure 5.

**Reviewer #2 (Public review):**
This is an interesting study exploring methods for reconstructing visual stimuli from neural activity in the mouse visual cortex. Specifically, it uses a competition dataset (published in the Dynamic Sensorium benchmark study) and a recent winning model architecture (DNEM, dynamic neural encoding model) to recover visual information stored in ensembles of the mouse visual cortex.This is a great project - the physiological data were measured at a single-cell resolution, the movies were reasonably naturalistic and representative of the real world, the study did not ignore important correlates such as eye position and pupil diameter, and of course, the reconstruction quality exceeded anything achieved by previous studies. Overall, it is great that teams are working towards exploring image reconstruction. Arguably, reconstruction may serve as an endgame method for examining the information content within neuronal ensembles - an alternative to training interminable numbers of supervised classifiers, as has been done in other studies. Put differently, if a reconstruction recovers a lot of visual features (maybe most of them), then it tells us a lot about what the visual brain is trying to do: to keep as much information as possible about the natural world in which its internal motor circuits may act consequently.While we enjoyed reading the manuscript, we admit that the overall advance was in the range of those that one finds in a great machine learning conference proceedings paper. More specifically, we found no major technical flaws in the study, only a few potential major confounds (which should be addressable with new analyses), and the manuscript did not make claims that were not supported by its findings, yet the specific conceptual advance and significance seemed modest. Below, we will go through some of the claims, and ask about their potential significance.

We thank the reviewer for the positive feedback on our paper.

(1) The study showed that it could achieve high-quality video reconstructions from mouse visual cortex activity using a neural encoding model (DNEM), recovering 10-second video sequences and approaching a two-fold improvement in pixel-by-pixel correlation over attempts. As a reader, I am left with the question: okay, does this mean that we should all switch to DNEM for our investigations of the mouse visual cortex? What makes this encoding model special? It is introduced as "a winning model of the Sensorium 2023 competition which achieved a score of 0.301... single-trial correlation between predicted and ground truth neuronal activity," but as someone who does not follow this competition (most eLife readers are not likely to do so, either), I do not know how to gauge my response. Is this impressive? What is the best achievable score, in theory, given data noise? Is the model inspired by the mouse brain in terms of mechanisms or architecture, or was it optimized to win the competition by overfitting it to the nuances of the data set? Of course, I know that as a reader, I am invited to read the references, but the study would stand better on its own if clarified how its findings depended on this model.

This is a very good point. We do not think that everyone should switch to using this particular DNEM to investigate the mouse visual cortex, but we think DNEMs and stimulus reconstruction in general has a lot of potential. We think static neural encoding models have already been demonstrated to be an extremely valuable tool to investigate visual coding (Walker et al., 2019; Yoshida et al., 2021; Willeke et al., bioRxiv 2023). DNEMs are less common, largely because they are very large and are technically more demanding to train and use. That makes static encoding models more practical for some applications, but they do not have temporal kernels and are therefore only used for static stimuli. They cannot, for instance, encode direction tuning, only orientation tuning. But both static and dynamic encoding models have advantages over stimulus classification methods which we outline in our discussion. Here we provide the first demonstration that previous achievements in static image reconstruction are transferable to movies.

It has been shown in the past for static neural encoding models that choosing a better-performing model produces reconstructed static images that are closer to the original image (Pierzchlewicz et al., 2023). The factors in choosing this particular DNEM were its capacity to predict neural activity (benchmarked against other models), it was open source, and the data it was designed for was also available.

To give more context to the model used in the paper, we have included the following, line 348:

“This model achieved an average single-trial correlation between predicted and ground truth neural activity of 0.291 during the competition, this was later improved to 0.301. The competition benchmark models achieved 0.106, 0.164 and 0.197 single-trial correlation, while the third and second place models achieved 0.243 and 0.265. Across the models, a variety of architectural components were used, including 2D and 3D convolutional layers, recurrent layers, and transformers, to name just a few.”

Concerning biologically inspired model design. The winning model contained 3 fully connected layers comprising the “Cortex” just before the final readout of neural activity, but we would consider this level of biological inspiration as minor. We do not think that the exact architecture of the model is particularly important, as the crucial aspect of such neural encoders is their ability to predict neural activity irrespective of how they achieve it. There has been a move towards creating foundation models of the brain (Wang et al., 2025) and the priority so far has been on predictive performance over mechanistic interpretability or similarity to biological structures and processes.

Finally, we would like to note that we do not know what the maximum theoretical score for single-trial responses might be, and don't think there is a good way of estimating it in this context.

(2) Along those lines, two major conclusions were that "critical for high-quality reconstructions are the number of neurons in the dataset and the use of model ensembling." If true, then these principles should be applicable to networks with different architectures. How well can they do with other network types?

This is a good question. Our method critically relies on the accurate prediction of neural activity in response to new videos. It is therefore expected that a model that better predicts neural responses to stimuli will also be better at reconstructing those stimuli given population activity. This was previously shown for static images (Pierzchlewicz et al., 2023). It is also expected that whenever the neural activity is accurately predicted, the corresponding reconstructed frames will also be more similar to the ground truth frames. We have now demonstrated this relationship between prediction accuracy and reconstruction accuracy in supplementary figure 4.

Although it would be interesting to compare the movie reconstruction performance of many different models with different architectures and activity prediction performances, this would involve quite substantial additional work because movie reconstruction is very resource- and time-intensive. Finding optimal hyperparameters to make such a comparison fair and informative would therefore be impractical and likely not yield surprising results.

We also think it is unlikely that ensembling would not improve reconstruction performance in other models because ensembling across model predictions is a common way of improving single-model performance in machine learning. Likewise, we think it is unlikely that the relationship between neural population size and reconstruction performance would differ substantially when using different models, because using more neurons means that a larger population of noisy neurons is “voting” on what the stimulus is. However, we would expect that if the model were worse at predicting neural activity, then more neurons are needed for an equivalent reconstruction performance. In general, we would recommend choosing the best possible DNEM available, in terms of neural activity prediction performance, when reconstructing movies using input optimization through gradient descent.

(3) One major claim was that the quality of the reconstructions depended on the number of neurons in the dataset. There were approximately 8000 neurons recorded per mouse. The correlation difference between the reconstruction achieved by 1 neuron and 8000 neurons was ~0.2. Is that a lot or a little? One might hypothesize that ~7,999 additional neurons could contribute more information, but perhaps, those neurons were redundant if their receptive fields were too close together or if they had the same orientation or spatiotemporal tuning. How correlated were these neurons in response to a given movie? Why did so many neurons offer such a limited increase in correlation?

In the population ablation experiments, we compared the performance using ~1000, ~2000, ~4000, ~8000 neurons, and found an attenuation of 39.5% in video correlation when dropping 87.5% of the neurons (~1000 neurons remaining), we did not try reconstruction using just 1 neuron.

(4) On a related note, the authors address the confound of RF location and extent. The study resorted to the use of a mask on the image during reconstruction, applied during training and evaluation (Line 87). The mask depends on pixels that contribute to the accurate prediction of neuronal activity. The problem for me is that it reads as if the RF/mask estimate was obtained during the very same process of reconstruction optimization, which could be considered a form of double-dipping (see the "Dead salmon" article, https://doi.org/10.1016/S1053-8119(09)71202-9). This could inflate the reconstruction estimate. My concern would be ameliorated if the mask was obtained using a held-out set of movies or image presentations; further, the mask should shift with eye position, if it indeed corresponded to the "collective receptive field of the neural population." Ideally, the team would also provide the characteristics of these putative RFs, such as their weight and spatial distribution, and whether they matched the biological receptive fields of the neurons (if measured independently).

We can reassure the reviewer that there is no double-dipping. We would like to clarify that the mask was trained only on videos from the training set of the DNEM and not the videos which were reconstructed. We have added the sentence, line 91:

“None of the reconstructed movies were used in the optimization of this transparency mask.”

Making the mask dependent on eye position would be difficult to implement with the current DNEM, where eye position is fed to the model as an additional channel. When using a model where the image is first transformed into retinotopic coordinates in an eye position-dependent manner (such as in Wang et al., 2025) the mask could be applied in retinotopic coordinates and therefore be dependent on eye position.

Effectively, the alpha mask defines the relative level of influence each pixel contributes to neural activity prediction. We agree it is useful to compare the shape of the alpha mask with the location of traditional on-off receptive fields (RFs) to clarify what the alpha mask represents and characterise the neural population available for our reconstructions. We therefore presented the DNEM with on-off patches to map the receptive fields of single neurons in an *in silico* experiment (the experimentally derived RF are not available). As expected, there is a rough overlap between the alpha mask (Supplementary Figure 2D), the average population receptive field (Supplementary Figure 2B), and the location of receptive field peaks (Supplementary Figure 2C). In principle, all three could be used during training or evaluation for masking, but we think that defining a mask based on the general influence of images on neural activity, rather than just on off patch responses, is a more elegant solution.

One idea of how to go a step further would be to first set the alpha mask threshold during training based on the % loss of neural activity prediction performance that threshold induces (in our case alpha=0.5 corresponds to ~3% loss in correlation between predicted vs recorded neural responses, see Supplementary Figure 3D), and second base the evaluation mask on a pixel correlation threshold (see example pixel correlation map in Supplementary Figure 2E) instead to avoid evaluating areas of the image with low image reconstruction confidence.

We referred to this figure in the result section, line 83:

“The transparency masks are aligned with but not identical to the On-Off receptive field distribution maps using sparse-noise (Figure S2).”

We have also done additional analysis on the effect of masking during training and evaluation with different thresholds in Supplementary Figure 3.

(5) We appreciated the experiments testing the capacity of the reconstruction process, by using synthetic stimuli created under a Gaussian process in a noise-free way. But this further raised questions: what is the theoretical capability for the reconstruction of this processing pipeline, as a whole? Is 0.563 the best that one could achieve given the noisiness and/or neuron count of the Sensorium project? What if the team applied the pipeline to reconstruct the activity of a given artificial neural network's layer (e.g., some ResNet convolutional layer), using hidden units as proxies for neuronal calcium activity?

That’s a very interesting point. It is very hard to know what the theoretical best reconstruction performance of the model would be. Reconstruction performance could be decreased due to neural variability, experimental noise, the temporal kernel of the calcium indicator and the imaging frame rate, information compression along the visual hierarchy, visual processing phenomena (such as predictive coding and selective attention), failure of the model to predict neural activity correctly, or failure of the reconstruction process to find the best possible image which explains the neural activity. We don't think we can disentangle the contribution of all these sources, but we can provide a theoretical maximum assuming that the model and the reconstruction process are optimal. To that end, we performed additional simulations and reconstructed the natural videos using the predicted activity of the neurons in response to the natural videos as the target (similar to the synthetic stimuli) and got a correlation of 0.766. So, the single trial performance of 0.569 is ~75% of this theoretical maximum. This difference can be interpreted as a combination of the losses due to neuronal variability, measurement noise, and actual deviations in the images represented by the brain compared to reality.

We thank the reviewer for this suggestion, as it gave us the idea of looking at error maps (Figure 6), where the pixel-level deviation of the reconstructions from recorded vs predicted activity is overlaid on the ground truth movie.

(6) As the authors mentioned, this reconstruction method provided a more accurate way to investigate how neurons process visual information. However, this method consisted of two parts: one was the state-of-the-art (SOTA) dynamic neural encoding model (DNEM), which predicts neuronal activity from the input video, and the other part reconstructed the video to produce a response similar to the predicted neuronal activity. Therefore, the reconstructed video was related to neuronal activity through an intermediate model (i.e., SOTA DNEM). If one observes a failure in reconstructing certain visual features of the video (for example, high-spatial frequency details), the reader does not know whether this failure was due to a lack of information in the neural code itself or a failure of the neuronal model to capture this information from the neural code (assuming a perfect reconstruction process). Could the authors address this by outlining the limitations of the SOTA DNEM encoding model and disentangling failures in the reconstruction from failures in the encoding model?

To test if a better neural prediction by the DNEM would result in better reconstructions, we ran additional simulations and now show that neural activity prediction performance correlates with reconstruction performance (Supplementary Figure 4B). This is consistent with Pierzchlewicz et al. (2023) who showed that static image reconstructions using better encoding models leads to better reconstruction performance. As also mentioned in the answer to the previous comment, untangling the relative contributions of reconstruction losses is hard, but we think that improvements to the DNEM performance are key. Two suggestions to improving the DNEM we used would be to translate the input image into retinotopic coordinates and shift this image relative to eye position before passing it to the first convolutional layer (as is done in Wang et al. 2025), to use movies which are not spatially down sampled as heavily, to not use a dilation of 2 in the temporal convolution of the first layer and to train on a larger dataset.

(7) The authors mentioned that a key factor in achieving high-quality reconstructions was model assembling. However, this averaging acts as a form of smoothing, which reduces the reconstruction's acuity and may limit the high-frequency content of the videos (as mentioned in the manuscript). This averaging constrains the tool's capacity to assess how visual neurons process the low-frequency content of visual input. Perhaps the authors could elaborate on potential approaches to address this limitation, given the critical importance of high-frequency visual features for our visual perception.

This is exactly what we also thought. To answer this point more specifically, we ran additional simulations where we also reconstruct the movies using gradient ensembling instead of reconstruction ensembling. Here, the gradients of the loss with respect to each pixel of the movie is calculated for each of the model instances and are averaged at every iteration of the reconstruction optimization. In essence, this means that one reconstruction solution is found, and the averaging across reconstructions, which could degrade high-frequency content, is skipped. The reconstructions from both methods look very similar, and the video correlation is, if anything, slightly worse (Supplemental Figure 3A&C). This indicates that our original ensembling approach did not limit reconstruction performance, but that both approaches can be used, depending on what is more convenient given hardware restrictions.

**Reviewer #3 (Public review):**
Summary:This paper presents a method for reconstructing input videos shown to a mouse from the simultaneously recorded visual cortex activity (two-photon calcium imaging data). The publicly available experimental dataset is taken from a recent brain-encoding challenge, and the (publicly available) neural network model that serves to reconstruct the videos is the winning model from that challenge (by distinct authors). The present study applies gradient-based input optimization by backpropagating the brain-encoding error through this selected model (a method that has been proposed in the past, with other datasets). The main contribution of the paper is, therefore, the choice of applying this existing method to this specific dataset with this specific neural network model. The quantitative results appear to go beyond previous attempts at video input reconstruction (although measured with distinct datasets). The conclusions have potential practical interest for the field of brain decoding, and theoretical interest for possible future uses in functional brain exploration.Strengths:The authors use a validated optimization method on a recent large-scale dataset, with a state-of-the-art brain encoding model. The use of an ensemble of 7 distinct model instances (trained on distinct subsets of the dataset, with distinct random initializations) significantly improves the reconstructions. The exploration of the relation between reconstruction quality and the number of recorded neurons will be useful to those planning future experiments.Weaknesses:The main contribution is methodological, and the methodology combines pre-existing components without any new original components.

We thank the reviewer for taking the time to review our paper and for their overall positive assessment. We would like to emphasise that combining pre-existing machine learning techniques to achieve top results in a new modality does require iteration and innovation. While gradient-based input optimization by backpropagating the brain-encoding error through a neural encoding model has been used in 2D static image optimization to generate maximally exciting images and reconstruct static images, we are the first to have applied it to movies which required accounting for the time domain. Previous methods used time averaged responses and were limited to the reconstruction of static images presented with fixed image intervals.

The movie reconstructions include a learned "transparency mask" to concentrate on the most informative area of the frame; it is not clear how this choice impacts the comparison with prior experiments. Did they all employ this same strategy? If not, shouldn't the quantitative results also be reported without masking, for a fair comparison?

Yes, absolutely. All reconstruction approaches limit the field of view in some way, whether this is due to the size of the screen, the size of the image on the screen, or cropping of the presented/reconstructed images during analysis due to the retinotopic coverage of the recorded neurons. Note that we reconstruct a larger field of view than Yoshida et al. In Yoshida et al., the reconstructed field of view was 43 by 43 retinal degrees. we show the size of an example evaluation mask in comparison.

To address the reviewer’s concern more specifically, we performed additional simulations and now also show the performance using a variety of different training and evaluation masks, including different alpha thresholds for training and evaluation masks as well as the effective retinotopic coverage at different alpha thresholds. Despite these comparisons, we would also like to highlight that the comparison to the benchmark is problematic itself. This is because image and movie reconstruction are not directly comparable. It does not make sense to train and apply a dynamic model on a static image dataset where neural activity is time averaged. Conversely, it does not make sense to train or apply a static model that expects time-averaged neural responses on continuous neural activity unless it is substantially augmented to incorporate temporal dynamics, which in turn would make it a new method. This puts us in the awkward position of being expected to compare our video reconstruction performance to previous image reconstruction methods without a fair way of doing so. We have therefore de-emphasised the phrasing comparing our method to previous publications in the abstract, results, and discussion.

Abstract: “We achieve a ~2-fold increase in pixel-by-pixel correlation compared to previous state-of-the-art reconstructions of static images from mouse V1, while also capturing temporal dynamics.” with “We achieve a pixel-level correction of 0.57 between the ground truth movie and the reconstructions from single-trial neural responses.”

Results: “This represents a ~2x higher pixel-level correlation over previous single-trial static image reconstructions from V1 in awake mice (image correlation 0.238 +/- 0.054 s.e.m. for awake mice) [Yoshida et al., 2020] over a similar retinotopic area (~43° x 43°) while also capturing temporal dynamics. However, we would like to stress that directly comparing static image reconstruction methods with movie reconstruction approaches is fundamentally problematic, as they rely on different data types both during training and evaluation (temporally averaged vs continuous neural activity, images flashed at fixed intervals vs continuous movies).”

Discussion: “In conclusion, we reconstruct videos presented to mice based on the activity of neurons in the mouse visual cortex, with a ~2-fold improvement in pixel-by-pixel correlation compared to previous static image reconstruction methods.” with “In conclusion, we reconstruct videos presented to mice based on single-trial activity of neurons in the mouse visual cortex.”

We have also removed the performance table and have instead added supplementary figure 3 with in-depth comparison across different versions of our reconstruction method (variations of masking, ensembling, contrast & luminance matching, and Gaussian blurring).

We believe that we have given enough information in our paper now so that readers can make an informed decision whether our movie reconstruction method is appropriate for the questions they are interested in.

**Recommendations for the authors:**

**Reviewer #2 (Recommendations for the authors):**
(1) "Reconstructions have been luminance (mean pixel value across video) and contrast (standard deviation of pixel values across video) matched to ground truth." This was not clear: was it done by the investigating team? I imagine that one of the most easily captured visual features is luminance and contrast, why wouldn't the optimization titrate these well?

The contrast and luminance matching of the reconstructions to the ground truth videos was done by us, but this was only done to help readers assess the quality of the reconstructions by eye. Our performance metrics (frame and video correlation) are contrast and luminance insensitive. To clarify this, we have also added examples of non-adjusted frames in Supplementary Figure 3A, and added a sentence in the results, line 103:

“When presenting videos in this paper we normalize the mean and standard deviation of the reconstructions to the average and standard deviation of the corresponding ground truth movie before applying the evaluation masks, but this is not done for quantification except in Supplementary Figure 3D.”

We were also initially surprised that contrast and luminance are not captured well by our reconstruction method, but this makes sense as V1 is largely luminance invariant (O’Shea et al., 2025 https://doi.org/10.1016/j.celrep.2024.115217) and contrast only has a gain effect on V1 activity (Tring et al., 2024 here). Decoding absolute contrast is likely unreliable because it is probably not the only factor modulating the overall gain of the neural population.

To address the reviewer’s comment more fully, we ran additional experiments. More specifically, to test why contrast and luminance are not recovered in the reconstructions, we checked how the predicted activity between the reconstruction and the contrast/luminance corrected reconstructions differs. Contrast and luminance adjustment had little impact on predicted response similarity on average. This makes the reconstruction optimization loss function insensitive to overall contrast and luminance so it cannot be decoded. There is a small effect on activity correlation, however, so we cannot completely rule out that contrast and luminance could be reconstructed with a different loss function.

(2) The authors attempted to investigate the variability in reconstruction quality across different movies and 10-second snippets of a movie by correlating various visual features, such as video motion energy, contrast, luminance, and behavioral factors like running speed, pupil diameter, and eye movement, with reconstruction success. However, it would also be beneficial if the authors correlated the response loss (Poisson loss between neural responses) with reconstruction quality (video correlation) for individual videos, as these metrics are expected to be correlated if the reconstruction captures neural variance.

We thank the reviewer for this suggestion. We have now included this analysis and find that if the neural activity was better predicted by the DNEM then the reconstruction of the video was also more similar to the ground truth video. We further found that this effect is shift-dependent (in time), meaning the prediction of activity based on proximal video frames is more influential on reconstruction performance.

**Reviewer #3 (Recommendations for the authors):**
(1) I was confused about the choice of applying a transparency mask thresholded with alpha>0.5 during training and alpha>1 during evaluation. Why treat the two situations differently? Also, shouldn't we expect alpha to be in the [0,1] range, in which case, what is the meaning of alpha>1? (And finally, as already described in "Weaknesses", how does this choice impact the comparison with prior experiments? Did they also employ a similar masking strategy?)

We found that applying a mask during training increased performance regardless of the size of the evaluation mask. Using a less stringent mask during training than during evaluation increases performance slightly, but also allows inspection of the reconstruction in areas where the model will be less confident without sacrificing performance, if this is desired. The thresholds of 0.5 and 1 were chosen through trial and error, but the exact values do not hold intrinsic meaning. The alpha mask values can go above 1 during their optimization. We could have clipped alpha during the training procedure (algorithm 1), but we decided this was not worth redoing at this stage, as the alphas used for testing were not above 1. All reconstruction approaches in previous publications limit the field of view in some form, whether this is due to the size of the screen, the size of the image on the screen, or the cropping of the presented/reconstructed images during analysis.

To address the reviewer’s comment in detail, we have added extensive additional analysis to evaluate the coverage of the reconstruction achieved in this paper and how different masking strategies affect performance, as well as how the mask relates to more traditional receptive field mapping.

(2) I would not use the word "imagery" in the first sentence of the abstract, because this might be interpreted by some readers as reconstruction of mental imagery, a very distinct question.

We changed imagery to images in the abstract.

(3) Line 145-146: "<1 frame, or <30Hz" should be "<1 frame, or >30Hz".

We have corrected the error.

(4) Algorithm 1, Line 5, a subscript variable 'g' should be changed to 'h'

We have corrected the error.

Additional Changes

(1) Minor grammatical errors

(2) Addition of citations: We were previously not aware of a bioRxiv preprint from 2022 (Cobos et al., 2022), which used gradient descent-based input optimization to reconstruct static images but without the addition of a diffusion model. Instead, we had cited for this method Pierzchlewicz et al., 2023 bioRxiv/NeurIPS. In Cobos et al., 2022, they compare static image reconstruction similarity to ground truth images and the similarity of the *in vivo* evoked activity across multiple reconstruction methods. Performance values are only given for reconstructions from trial-averaged responses across ~40 trials (in the absence of original data or code we are also not able to retrospectively calculate single-trial performance). The authors find that optimizing for evoked activity rather than image similarity produces image reconstructions that evoke more similar *in vivo* responses compared to reconstructions optimized for image similarity itself. We have now added and discussed the citation in the main text.

(3) Workaround for error in the open-source code from https://github.com/lRomul/sensorium for video hashing function in the SOTA DNEM: By checking the most correlated first frame for each reconstructed movie, we discovered there was a bug in the open-source code and 9/50 movies we originally used for reconstruction were not properly excluded from the training data between DNEM instances. The reason for this error was that some of the movies are different by only a few pixels, and the video hashing function used to split training and test set folds in the original DNEM code classified these movies as different and split them across folds. We have replaced these 9 movies and provide a figure below showing the next closest first frame for every movie clip we reconstruct. This does not affect our claims. Excluding these 9 movie clips, did not affect the reconstruction performance (video correlation went from 0.563 to 0.568), so there was no overestimation of performance due to test set contamination. However, they should still be removed so some of the values in the paper have changed slightly. The only statistical test that was affected was the correlation between video correlation and mean motion energy (Supplementary Figure 4A), which went from p = 0.043 to 0.071.

**Author response image 2. sa3fig2:** exclusion of movie clips with duplicates in the DNEM training data. A) example frame of a reconstructed movie (ground truth) and the most correlated first frame from the training data. b) all movie clips and their corresponding most correlated clip from the training data. Red boxes indicate excluded duplicates.